# MAST4 promotes primary ciliary resorption through phosphorylation of Tctex-1

Kensuke Sakaji[1], Sara Ebrahimiazar[1,2], Yasuhiro Harigae[1], Kenichi Ishibashi[3], Takeya Sato[1,4] , Takeo Yoshikawa[5,6], Gen-ichi Atsumi[3], Ching-Hwa Sung[7,8] , Masaki Saito[1,3]

**The primary cilium undergoes cell cycle–dependent assembly and disassembly. Dysregulated ciliary dynamics are associated with several pathological conditions called ciliopathies. Previous studies showed that the localization of phosphorylated Tctex-1 at Thr94 (T94) at the ciliary base critically regulates ciliary resorption by accelerating actin remodeling and ciliary pocket membrane endocytosis. Here, we show that microtubule-associated serine/threonine kinase family member 4 (MAST4) is localized at the primary cilium. Suppressing MAST4 blocks serum-induced ciliary resorption, and overexpressing MAST4 accelerates ciliary resorption. Tctex-1 binds to the kinase domain of MAST4, in which the R503 and D504 residues are key to MAST4-mediated ciliary resorption. The ciliary resorption and the ciliary base localization of phospho-(T94)Tctex-1 are blocked by the knockdown of MAST4 or the expression of the catalytic-inactive site-directed MAST4 mutants. Moreover, MAST4 is required for Cdc42 activation and Rab5-mediated periciliary membrane endocytosis during ciliary resorption. These results support that MAST4 is a novel kinase that regulates ciliary resorption by modulating the ciliary base localization of phospho-(T94)Tctex-1. MAST4 is a potential new target for treating ciliopathies causally by ciliary resorption defects.**

## Introduction

The primary cilium (9 + 0 architecture) is a microtubule-based immotile sensory organelle in vertebrates (Tucker et al, 1979). Enormous progress has been made in the past several decades concerning our understanding of the structure and functions of the primary cilium (Sung & Leroux, 2013). The subdomain between the ciliary axoneme and the basal body called the transition zone serves as a signal transduction hub and a barrier that modulates the trafficking of selected ciliary membrane proteins (Yeh et al, 2013; Schou et al, 2015; Goncalves & Pelletier, 2017; Hsiao et al, 2021). The ciliary pocket describes the invagination of the plasma membrane surrounding the proximal region of the primary cilium. This periciliary subdomain is enriched with clathrin coats and serves as an active site for clathrin-mediated endocytosis (Molla-Herman et al, 2010; Clement et al, 2013; Yeh et al, 2013; Saito et al, 2017). Ciliopathies are a class of diseases resulting from impaired ciliary structures and/or functions.

Dysregulated ciliary dynamics have been associated with several pathological conditions, including microcephaly and cancer (Li et al, 2011; Yeh et al, 2013; Gabriel et al, 2016; Chen et al, 2019; Zhang et al, 2019; Farooq et al, 2020; Shiromizu et al, 2020). Although the mechanism underlying the assembly has been extensively investigated, the mechanism and cellular machinery that underlies ciliary resorption are less well understood (Hsu et al, 2017; Saito et al, 2017; Wang & Dynlacht, 2018; Patel & Tsiokas, 2021). Ciliary dynamics have been widely studied in several non-transformed cell lines, including human retinal pigment epithelial cells (RPE-1), mouse embryonic fibroblasts, and NIH3T3 cells (Saito et al, 2018). These studies showed that serum withdrawal triggers quiescence and ciliogenesis, whereas serum readdition induces ciliary resorption. The kinetic studies showed that the cilium resorption is biphasic, happening 2 and 24 h after serum stimulation, which corresponds to the $G_1$-S and $G_2$-M transition, respectively (Pugacheva et al, 2007; Li et al, 2011; Saito et al, 2017). Like the serum, several other factors, such as insulin-like growth factor-1 (IGF-1) and platelet-derived growth factor, can also trigger ciliary resorption (Pugacheva et al, 2007; Li et al, 2011; Yeh et al, 2013; Hu et al, 2021).

The mechanism that underlies the cilium resorption was first studied by Pugacheva et al (2007). These authors showed that during ciliary resorption, ciliary axonemal microtubules are destabilized and depolymerized through the Aurora A kinase

[1]Department of Molecular Pharmacology, Tohoku University Graduate School of Medicine, Sendai, Japan   [2]Department of Developmental Neuroscience, Tohoku University Graduate School of Medicine, Sendai, Japan   [3]Department of Molecular Physiology and Pathology, School of Pharma-Science, Teikyo University, Itabashi-ku, Tokyo, Japan   [4]Department of Clinical Biology and Hormonal Regulation, Tohoku University Graduate School of Medicine, Sendai, Japan   [5]Department of Neuropharmacology, Hokkaido University Graduate School of Medicine, Sapporo, Japan   [6]Department of Pharmacology, Tohoku University Graduate School of Medicine, Sendai, Japan   [7]Department of Ophthalmology, Margaret M. Dyson Vision Research Institute, Weill Cornell Medicine, New York, NY, USA   [8]Department of Cell and Developmental Biology, Weill Cornell Medicine, New York, NY, USA

Correspondence: chsung@med.cornell.edu; saitou.masaki.nb@teikyo-u.ac.jp

(AurA)– and histone deacetylase 6 (HDAC6)–regulated pathway. Subsequent studies showed that the ciliary tip is decapitated by AurA (Phua et al, 2017). NIMA-related protein kinase 2 (Nek2) and polo-like kinase 1 (PLK1) also participate in the AurA-dependent pathway during ciliary resorption (Lee et al, 2012; Spalluto et al, 2012). Preventing the ciliary (re)assembly, through the processes regulated by the cyclin-dependent kinase Cdk5, ubiquitin ligase FBW7, and centrosomal protein Nde1, further promotes the ciliary resorption (Kim et al, 2011; Maskey et al, 2015).

We have previously identified Tctex-1, or dynein light-chain Tctex-type 1 (DYNLT1), as a vital component for ciliary resorption (Li et al, 2011). Tctex-1 has both dynein-dependent and dynein-independent roles (Chuang et al, 2005; Liu et al, 2015). Our previous studies showed that Tctex-1, upon phosphorylation at Thr94 (i.e., phospho-(T94)Tctex-1), is enriched in the ciliary transition zone of cells at the $G_1$-S border and promotes ciliary resorption and $G_1$-S progression (Li et al, 2011). Knockdown (KD) of Tctex-1 induced the coupled defects in ciliary resorption and cell cycle re-entry. These Tctex-1 suppression–caused defects can be rescued by the expression of either WT Tctex-1 or the phospho-mimetic variant Tctex-$1^{T94E}$, but not the phospho-dead variant Tctex-$1^{T94A}$. Mechanistically, phospho-(T94)Tctex-1 regulates ciliary resorption through activating the Cdc42 small GTPase and its downstream effector subunit 2 of actin-related protein 2/3 (Arp2/3)–mediated actin branching, as well as Rab5-mediated endocytosis at the ciliary pocket membranes (Saito et al, 2017). To date, the mechanism underlying the phosphorylation of (T94)Tctex-1 remains unknown.

Microtubule-associated serine/threonine kinase family (MAST) belongs to the protein kinase A, G, and C (AGC) family of serine/threonine kinases. Four *MAST* family members (*MAST1-4*) are encoded by different genes. MAST4 was identified as one of the proteins that potentially interacted with Tctex-1 during the previous proteomics studies (Saito et al, 2017) (also see below). MAST4 is a large protein (~265 kD); it comprises 2,434 amino acids and encompasses a DUF1908, a serine/threonine kinase, and a post-synaptic density-95/disks large/zonula occludens-1 (PDZ) domain. MAST4 has several reported functions in neuroprotection, spermatogenesis, the progression of multiple myeloma bone disease, and cell fate determination of mesenchymal stromal cells in bones and cartilages (Gongol et al, 2017; Lee et al, 2021; Cui et al, 2022; Kim et al, 2022). In this study, we describe the ciliary localization of MAST4 and the physical and functional interactions between MAST4 and Tctex-1 in ciliary resorption.

# Results

## MAST4 positively regulates ciliary resorption

MAST4 is one of the proteins pulled down alongside Tctex-1 from 2-h post–serum-stimulated RPE-1 cells that stably expressed a Tctex-1-GFP fusion (Saito et al, 2017) (see Table S1). Quantitative real-time PCR (qRT-PCR) showed that MAST4 is the predominant MAST family member expressed in RPE-1 cells (Fig S1). Immunoblotting assay showed that an endogenous MAST4 protein was greatly reduced by the transfection of two short hairpin RNAs (sh1 and sh2) that targeted independent sequences (Fig S2A).

We first investigated the role played by MAST4 in ciliary resorption using the KD approach. We subjected MAST4-sh1– and MAST4-sh2–transfected RPE-1 cells to ciliary resorption assays, that is, inducing the ciliogenesis by serum starvation and then adding the serum back. The MAST4-shRNA plasmids also expressed GFP via internal ribosome entry site (IRES) so that the MAST4-KD cells could be identified by the expression of GFP. Confocal microscopic examination of acetylated $\alpha$-tubulin (Ac-Tub)–labeled cilium was used to determine the length of cilia in the GFP+-transfected cells and the percentage of GFP+-ciliated cells. Both measurements showed, as expected (Pugacheva et al, 2007; Li et al, 2011; Saito et al, 2017), that control cells expressing GFP alone underwent biphasic resorption at 2 and 24 h after serum readdition (Fig 1A–C). In contrast, MAST4-shRNA transfection abrogated ciliary resorption at both 2- and 24-h time points (Fig 1A–C). MAST4-sh2 slightly affected the ciliary assembly. Therefore, we used MAST4-sh1 for the subsequent studies to avoid any possible complication in data interpretation (Fig 1A–C).

We then asked whether MAST4 overexpression promotes ciliary resorption. Immunoblots showed that MAST4 (MAST4$^{WT}$) expressed in RPE-1 cells migrated according to the expected molecular weight (Fig S2B). Unlike the control cells in which ciliary resorption began 2 h post-serum readdition, the MAST4$^{WT}$-transfected cells had ciliary resorption that began 30 min after serum readdition (Fig 1D). The percentage of ciliated cells continued to reduce in these cells at the later time points (1–24 h).

Our pilot studies showed that none of the commercial MAST4 antibodies was useful for immunostaining the endogenous MAST4. We thus examined the possible ciliary distribution of MAST4 using the transiently transfected mCherry fusion of MAST4$^{WT}$ (MAST4$^{WT}$-mCherry). In quiescent RPE-1 cells, the MAST4$^{WT}$-mCherry signal was overlapped with the $\gamma$-tubulin–labeled basal body, the Ac-Tub–labeled ciliary axoneme, and the space in between (Fig 1E). Super-resolution Airyscan examination provided additional evidence supporting the ciliary localization of MAST4$^{WT}$-mCherry (Fig S3). These findings collectively support the idea that the cilium-localized MAST4 plays an active role in ciliary resorption.

## Tctex-1 binds to the kinase domain of MAST4

We set out to experimentally confirm the interaction between MAST4 and Tctex-1. First, we showed that both MAST4$^{WT}$-mCherry and FLAG-tagged Tctex-1 co-expressed in the HEK293 cells can be immunoprecipitated by anti-mCherry antibody, but not by the control anti-GFP antibody (Fig 2A).

Next, we characterized the protein–protein interaction by mapping the motif(s) of MAST4 that bind(s) to Tctex-1. We employed pull-down assays using bacterially expressed recombinant proteins. We generated GST fusion proteins containing the DUF1908 domain (fragment 1), the serine/threonine kinase domain (fragment 2), the PDZ domain (fragment 3), or several distinct parts of the remaining region of MAST4 (fragments 4–7) (Fig 2B). Bacterial lysates containing roughly an equal amount of various GST-MAST4 fragments (Fig 2C, lower) were incubated with the lysates expressing a maltose-binding protein fusion of Tctex-1$^{WT}$ (MBP-Tctex-1$^{WT}$). The GST proteins bound to glutathione beads were eluted and subjected to electrophoresis and MBP immunoblotting assays. Although a weak signal of MBP-Tctex-1$^{WT}$ was pulled down by

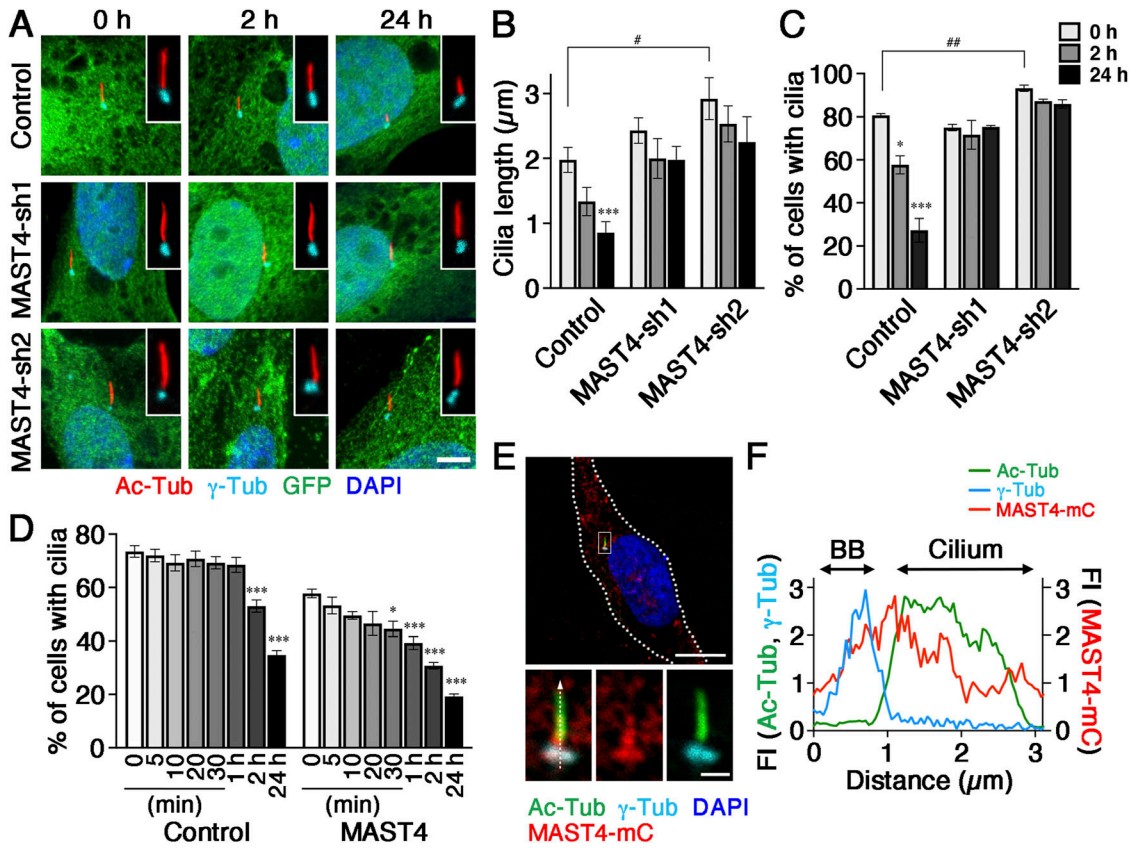

**Figure 1.  MAST4 is localized in the cilium and required for ciliary resorption.**
**(A)** RPE-1 cells transfected with the GFP alone (control), MAST4-shRNA1-IRES-GFP (MAST4-sh1), or MAST4-shRNA2-IRES-GFP (MAST4-sh2) were starved and added serum back for the indicated time periods. Representative confocal images of the Ac-Tub–labeled cilium (red) and the γ-Tub–labeled basal body (cyan) in GFP⁺ cells (green) are shown. Cell nuclei were stained with DAPI (blue). $n$ = 16–31 cells from three independent experiments. Scale bar: 5 $\mu m$. **(B, C)** Quantification of (A). Histogram shows the cilium length (measured by Ac-Tub) (B) and the fraction of the GFP⁺ cells expressing a cilium (C). Values are means ± S.E.M. *$P$ < 0.05 and ***$P$ < 0.001, one-way ANOVA followed by Tukey's test versus 0 h for each group. #$P$ < 0.05 and ##$P$ < 0.01, two-way ANOVA followed by Bonferroni's test. $n$ = 16–31 cells (B) and $n$ = 100 cells in each experiment, with three independent experiments (C). **(D)** RPE-1 cells transiently transfected with GFP (control) or GFP together with MAST4^WT were starved and followed by serum readdition. The quantification of the percentage of GFP⁺ cells, harvested at different time points post-serum stimulation, is shown. Values are means ± S.E.M. *$P$ < 0.05 and ***$P$ < 0.001, one-way ANOVA followed by Tukey's test versus 0 h for each group. $n$ = 100 cells in each experiment, with three independent experiments. **(E)** Representative confocal images of MAST4-mCherry (mC) in quiescent transfected RPE-1 cells stained for mCherry (red), Ac-Tub (green), and γ-Tub (cyan). White dashed lines demarcate the cell border (DAPI: nuclear). The high-power view highlights the ciliary distribution of MAST4-mCherry. Scale bars: 10 $\mu m$ (E, low-power magnification) and 1 $\mu m$ (E, high-power magnification). **(E, F)** Segmented profiles of fluorescence intensities (FIs) along the white dashed arrow shown in (E). BB, basal body. The y-axis FI unit is defined by the acquired values divided by 10⁴ using Zen software (Zeiss). The x-axis describes the distance from the proximal part of the basal body (considered as 0). The representative line scan from the images of 16 cilia is shown.

fragment 1, MBP-Tctex-1^WT was pulled down the most by fragment 2 (Fig 2C, upper). These results suggest that Tctex-1 interacts with the kinase domain of MAST4.

## MAST4 is important for the ciliary base activation of phospho-(T94)Tctex-1

The above studies prompted us to hypothesize that MAST4 is involved in the phosphorylation of Tctex-1 and, in turn, controls the phospho-(T94)Tctex-1–mediated ciliary resorption. Toward testing this model, we performed ciliary resorption assays in RPE-1 cells expressing a MAST4-sh1 plasmid together with FLAG-Tctex-1^WT, Tctex-1^T94E, or Tctex-1^T94A. As shown in Fig 3A, the ciliary resorption blocked by MAST4-KD was significantly rescued by the co-expression of Tctex-1^T94E. In contrast, Tctex-1^WT and Tctex-1^T94A failed to rescue. The converse experiments showed the accelerated

ciliary resorption caused by MAST4^WT overexpression was blunted by the co-expression of FLAG-Tctex-1^T94A (Fig 3B).

Previous studies showed that phospho-(T94)Tctex-1 prominently appeared at the ciliary base 2 h post-serum readdition (Li et al, 2011). We showed that compared with the control, MAST4-suppressed cells had significantly less serum-induced phospho-(T94)Tctex-1 signal at the ciliary base (Fig 3C and D). These results collectively suggest that MAST4 acts upstream of the phospho-(T94)Tctex-1–mediated ciliary resorption through the ciliary base localization of phospho-(T94) Tctex-1.

## The catalytic motif of MAST4 is important for Tctex-1–mediated ciliary resorption

Having identified the kinase domain (i.e., catalytic motif) as the Tctex-1–binding motif, we went ahead to determine which residue(s) in this

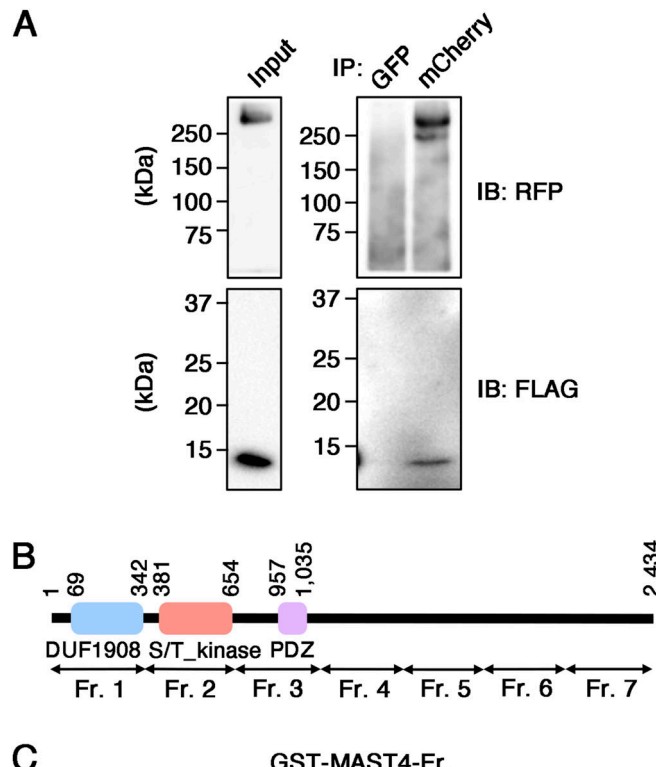

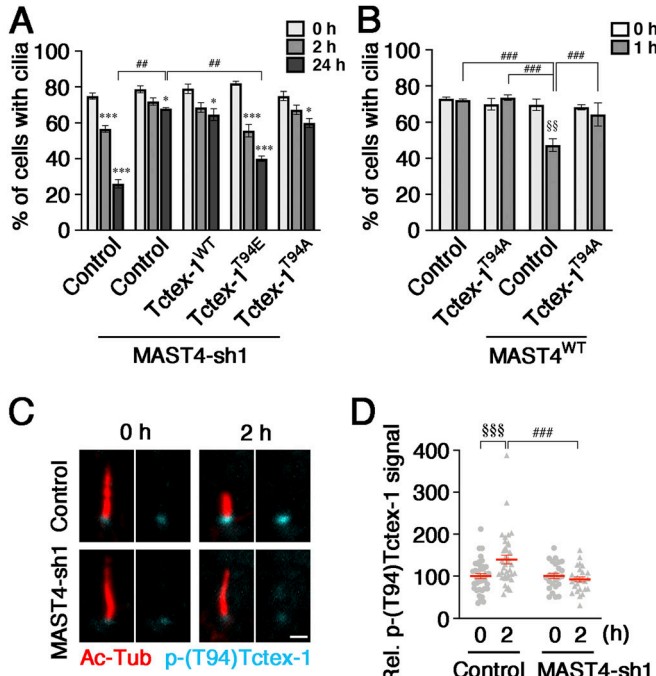

**Figure 3. MAST4 regulates Tctex-1–mediated ciliary resorption and the appearance of phospho-(T94)Tctex-1 at the ciliary base.**
**(A)** RPE-1 cells transfected with GFP alone (control), or MAST4-sh1-IRES-GFP (MAST4-sh1) together with FLAG-tagged Tctex-1$^{WT}$, Tctex-1$^{T94E}$, or Tctex-1$^{T94A}$ were subjected to a ciliary resorption assay. The histogram shows the percentage of GFP$^+$ cells expressing Ac-Tub–labeled cilium at the indicated time points post-serum restimulation. Values are means ± S.E.M. *$P < 0.05$ and ***$P < 0.001$, one-way ANOVA followed by Tukey's test versus 0 h for each group. $^{##}P < 0.01$, two-way ANOVA followed by Bonferroni's test. $n = 100$ cells in each experiment, with three independent experiments. **(B)** Ciliary resorption assay of RPE-1 cells transfected with FLAG-Tctex-1$^{T94A}$ with or without co-transfected MAST4$^{WT}$. The histogram shows the percentage of ciliated GFP$^+$ cells at 0 and 1 h post-serum readdition. Values are means ± S.E.M. $^{###}P < 0.001$, two-way ANOVA followed by Bonferroni's test. $^{§§}P < 0.01$, $t$ test versus 0 h. $n = 100$ cells in each experiment, with three independent experiments. **(C, D)** Representative confocal images of phospho-(T94)Tctex-1 (cyan) and Ac-Tub (red) in RPE-1 cells transfected with GFP (control) or MAST4-sh1 under the starving (0 h) and 2-h serum-stimulated conditions. **(C)** Phospho-(T94)Tctex-1 was detected by Alexa Fluor 647–conjugated secondary antibody and is pseudo-colored in cyan for presentation. **(D)** Fluorescence intensity of phospho-(T94)Tctex-1 associated with the proximal end of Ac-Tub–labeled cilium was quantified and is shown in (D). Scale bar: 1 μm (C). Values are means ± S.E.M. $^{###}P < 0.001$, two-way ANOVA followed by Bonferroni's test. $^{§§§}P < 0.001$, $t$ test. $n = 25–39$ cells from three independent experiments.

and human MAST2 (Johnson et al, 1996; Karpov et al, 2010). In PKA catalytic subunit α, R165 and D166 in the [HY]-[RY]-D-[LI]-K-P-[ED]-N motif (aka RD motif) and the D184, F185, and G186 residues (aka DFG motif) (Fig 4A) are critical for its kinase activity (Hanks & Hunter, 1995). R165 interacts with T197, which is autophosphorylated. D166 catalyzes and transfers the γ-phosphate of ATP to the serine/threonine residues in the substrates. D184 associates with Mg$^{2+}$ that binds to the β- and γ-phosphates of ATP. We thus predicted that R503 and D504 in human MAST4 constitute the RD motif, K506 is the proximate lysine residue, D522/F523/G524 is the DFG motif, and T534 or T535 is the autophosphorylation site (Fig 4A). The 3D structural modeling (AlphaFold2) is compatible with this prediction; R503, D504, and K506 are in a highly flexible module, which is favorable for the transfer of the γ-phosphate of ATP (Fig S4).

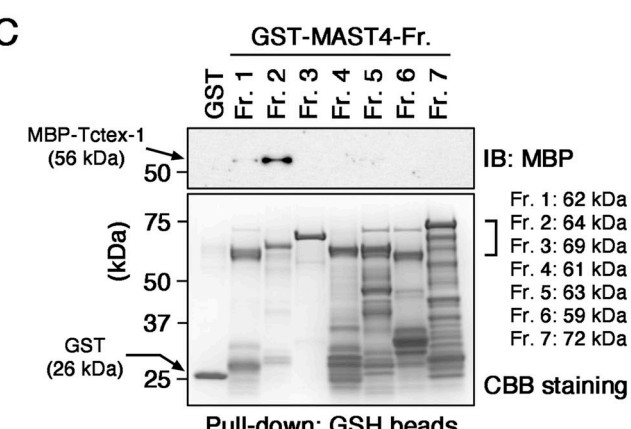

**Figure 2. Characterization of Tctex-1-MAST4 interaction and mapping Tctex-1–binding motif of MAST4.**
**(A)** Representative immunoblot of total HEK293 cell lysates containing transiently transfected MAST4$^{WT}$-mCherry and FLAG-Tctex-1$^{WT}$ (input) and the immunoprecipitants (IP) pulled down by anti-mCherry or anti-GFP (control) antibodies probed with anti-RFP and anti-DYKDDDDK (FLAG) antibodies. **(B)** Diagram showing different domains and corresponding amino acids of MAST4. Different regions (Fr.1–Fr. 7) of MAST4 fused with GST for pull-down assays are also labeled. S/T_kinase, serine/threonine kinase domain. **(C)** MBP-Tctex-1–containing bacterial lysates were mixed with bacterial extracts containing GST or GST-MAST4 fragments (with indicated molecular weights) and then incubated with glutathione beads. The eluates from the glutathione beads were electrophoresed and immunoblotted (IB) with anti-MBP antibody, or GST or GST-MAST4 fragments in the elutes were stained with Coomassie brilliant blue (CBB). The expected molecular weights of GST (26 kD) and GST-MAST4 are shown. Note the roughly equivalent amounts of GST and GST-MAST4 fragments were loaded in the inputs.

domain are required for ciliary resorption. We predicted potential residues central to the kinase activity of MAST4 based on the analogy with two other AGC family members, PKA catalytic subunit α

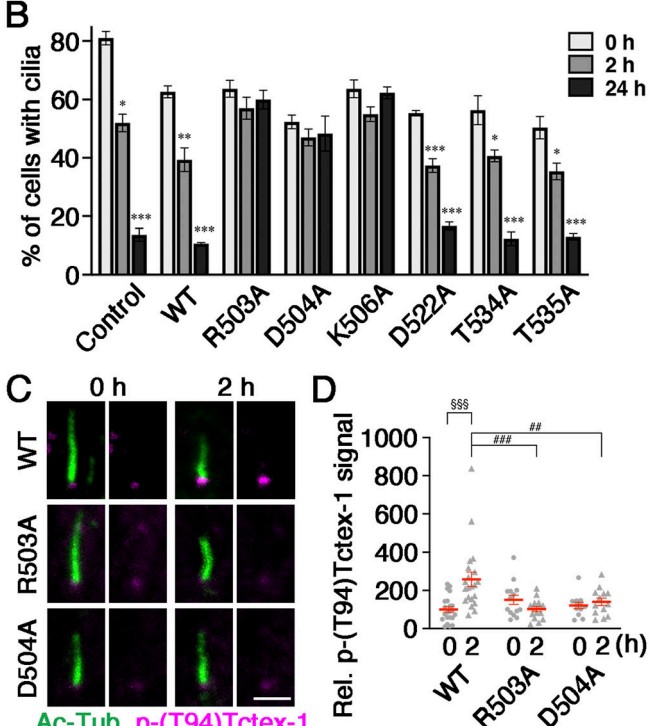

**Figure 4. Mapping catalytic motif residues of MAST4 required for the ciliary base activation of phospho-(T94)Tctex-1.**

**(A)** Alignment of the kinase domain of MAST2 and PKA catalytic subunit α (PKACA) with the predicted catalytic domain of MAST4. Bold letters were used to highlight the amino acid residues that were changed to alanine by site-directed mutagenesis. The predicted catalytic motif (H502–N509), DFG motif (D522–G524), and autophosphorylation residues (T524 or T535) of MAST4 are marked by a bar. **(B)** Ciliary resorption assay of RPE-1 cells transfected with GFP alone (control), or GFP together with MAST4[WT] or the indicated MAST variants. The histogram shows the percentage of ciliated GFP[+] cells at different time points after serum readdition. Values are means ± S.E.M. *P < 0.05, **P < 0.01, and ***P < 0.001, one-way ANOVA followed by Tukey's test versus 0 h for each group. n = 100 cells in each experiment, with three independent experiments. **(C)** Representative confocal images of phospho-(T94)Tctex-1 (magenta) and Ac-Tub (green) in RPE-1 cells transfected with GFP together with WT or indicated mutant MAST4-mCherry under the starving (0 h) and 2-h serum-stimulated conditions. **(C)** Phospho-(T94)Tctex-1 was detected by Alexa Fluor 647–conjugated secondary antibody and is pseudo-colored in magenta for presentation. n = 12–16 cells from three independent experiments. **(D)** Fluorescence intensity of phospho-(T94)Tctex-1 associated with the proximal end of Ac-Tub–labeled cilium was quantified (D). Values are means ± S.E.M. ##P < 0.01 and ###P < 0.001, two-way ANOVA followed by Bonferroni's test. §§§P < 0.001, t test. n = 12–16. Scale bar: 1 μm (C).

Subsequently, we made six site-directed mutants—MAST4[R503A], MAST4[D504A], MAST4[K506A], MAST4[D522A], MAST4[T534A], and MAST4[T535A]—by replacing R503, D504, K506, D522, T534, and T535 with alanine, respectively. To rule out any folding problem caused by the amino acid alteration, we confirmed all six variants were expressed as the expected size on immunoblots (Fig S5). Ciliary resorption assays showed that the transient transfection of MAST4[D522A], MAST4[T534A],

and MAST4[T535A] did not affect serum-induced ciliary resorption. In contrast, the transfection of MAST4[R503A], MAST4[D504A], and MAST4[K506A] significantly suppressed ciliary resorption at both 2- and 24-h time points post-serum readdition (Fig 4B). For unclear reasons, the transfection rate of MAST4[K506A] was lower than that of other variants; this was consistent with its weaker signal in immunoblots (Fig S5). Because we only counted the cilium expression in GFP[+]-transfected cells, the lower transfection rate of MAST4[K506A] would not affect the ciliary counting results. Nevertheless, for the subsequent studies, we primarily focused on testing MAST4[R503A] and MAST4[D504A] mutants.

We showed that the ciliary distribution of the MAST4[R503A] and MAST4[D504A] mutants was comparable to that of MAST4[WT] (Fig S6). This excludes the possibility that the adverse effects caused by these mutants were due to the mislocalization. Furthermore, we showed that the expression of MAST4[R503A] and MAST4[D504A] significantly reduced the intensity of the ciliary base signal of phospho-(T94)Tctex-1 at the 2-h time point (Fig 4C and D). These results, taken together, suggest that the R503 and D504 residues in the kinase domain of MAST4 are required for the serum-induced ciliary resorption, likely through modulating the ciliary base localization of phospho-(T94)Tctex-1.

### MAST4 regulates ciliary resorption by activating the downstream effectors of phospho-(T94)Tctex-1

Our previous studies showed that Cdc42-Arp2/3-modulated branched actin organization and Rab5-modulated periciliary membrane endocytosis act downstream to the phospho-(T94) Tctex-1–mediated ciliary resorption (Saito et al, 2017). We set out a serial experiment to demonstrate that Cdc42, Arp2/3, and Rab5 also act downstream to MAST4-mediated ciliary resorption. Cdc42 operates by alternating between an active, GTP-bound state, and an inactive, GDP-bound state (Ridley & Hall, 1992). Dominant negative (DN) Cdc42 variant, Cdc42[T17N], and constitutively active (CA) variant (Cdc42[G12V]) have been established (Feig & Cooper, 1988; Miller & Johnson, 1994). Ciliary resorption assays showed that the co-expression of the DN variant Cdc42[T17N] blocked MAST4[WT]-mediated ciliary resorption acceleration (Fig 5A). Conversely, the co-expression of the CA variant Cdc42[G12V] rescued the ciliary resorption inhibition caused by MAST4[R503A] and MAST4[D504A] (Fig 5B). Transfection of Cdc42-CA alone did not affect the ciliary resorption at 2- and 24-h time points (Fig 5B), as previously described (Saito et al, 2017).

Previous studies showed that serum addition in quiescent RPE-1 cells acutely promotes Cdc42 GTPase activity (i.e., at 1- and 2-h time points) (Saito et al, 2017). We measured the Cdc42 activity using a biochemical assay based on the rationale that the active form of Cdc42 binds to the p21-binding domain (PBD) of p21-activated kinase 1 (PAK) with a high affinity (Zhang et al, 1998). The amount of the activated Cdc42 pulled down by the PBD-PAK beads and the total Cdc42, both detected by the quantitative Cdc42 immunoblotting assays, were compared (Figs 5C–F and S7). These studies showed that KD of MAST4 significantly suppressed the Cdc42 activation at both 1- and 2-h time points post-serum stimulation (Figs 5C and D and S7A). The expression of MAST4[R503A] or MAST4[D504A] also significantly suppressed the Cdc42 activation at the 2-h time point

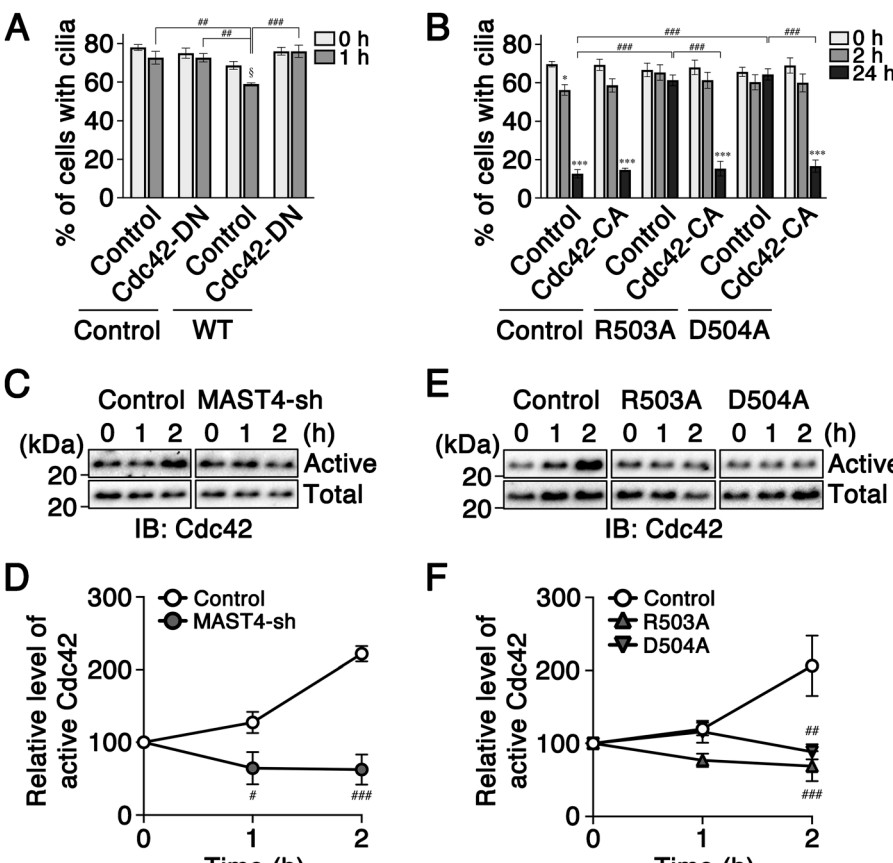

**Figure 5. MAST4 promotes ciliary resorption through Cdc42 activation.**
**(A)** Ciliary resorption assay of RPE-1 cells transfected with GFP alone (control), or MAST4$^{WT}$ together without (control) or with HA-tagged Cdc42$^{T17N}$ (Cdc42-DN). The histogram shows the percentage of GFP$^+$ cells with Ac-Tub–labeled primary cilia at the indicated time points. Values are means ± S.E.M. $^{##}P < 0.01$ and $^{###}P < 0.001$, two-way ANOVA followed by Bonferroni's test. $^{§}P < 0.05$, t test. $n$ = 100 cells in each experiment, with three independent experiments. **(B)** RPE-1 cells expressing GFP alone (control), MAST4$^{R503A}$, or MAST4$^{D504A}$ (co-expressed with GFP), in the presence or absence (control) of HA-tagged Cdc42$^{G12V}$ (Cdc42-CA). The cells were subjected to a ciliary resorption assay. The histogram shows the percentage of ciliated GFP$^+$ cells at different time points post–serum readdition. Values are means ± S.E.M. $^{*}P < 0.05$ and $^{***}P < 0.001$, one-way ANOVA followed by Tukey's test versus 0 h for each group. $^{###}P < 0.001$, two-way ANOVA followed by Bonferroni's test. $n$ = 100 cells in each experiment, with three independent experiments. **(C, D, E, F)** Cdc42 activity assays. **(C, E)** Representative Cdc42 immunoblots containing total Cdc42 and active Cdc42 pulled down by PAK-PBD beads from RPE-1 cells transfected with pCAG (control) and pCAG-MAST4-sh1 (C) or MAST4 variants (E). **(D, F)**. Ratios of active Cdc42 to total Cdc42 in cells harvested at different time points post–serum readdition are shown in (D, F). Value at the 0-h time point was considered as 100. Values (D, F) are means ± S.E.M. $^{#}P < 0.05$, $^{##}P < 0.01$, and $^{###}P < 0.001$, two-way ANOVA followed by Bonferroni's test. $n$ = 3 (D, F).

(Figs 5E and F and S7B). These results suggest that inhibiting the expression or function of MAST4 attenuates the serum-induced Cdc42 activation.

Furthermore, we showed that the suppression of the core subunit of the Arp2/3 complex, ARPC2, using a previously validated shRNA (Saito et al, 2017), abrogated the ciliary resorption acceleration caused by the MAST4$^{WT}$ overexpression (Fig S8).

The ciliary pocket is known as a hotspot for clathrin-mediated endocytosis (Molla-Herman et al, 2010; Saito et al, 2017). Serum (or IGF-1)-treated RPE-1 cells had the transferrins internalized through clathrin-mediated endocytosis concentrated near the ciliary pocket membranes (Molla-Herman et al, 2010; Clement et al, 2013; Yeh et al, 2013; Saito et al, 2017). Indeed, extracellularly added Alexa Fluor 594–conjugated transferrins were abundantly observed in the periciliary region in the control cells 1 h post–IGF-1 induction. The periciliary signal of Alexa Fluor 594–conjugated transferrins was largely reduced in MAST4-KD cells (Fig 6A). Moreover, we showed that the co-expression of the CA variant of Rab5 (Rab5$^{Q79L}$) significantly rescued the ciliary resorption inhibition caused by the co-transfected MAST4-sh1 (Fig 6B). Taken together, these results suggest that MAST4- and phospho-(T94)Tctex-1–mediated ciliary resorption use common downstream effectors.

## Discussion

Much progress has been made toward our understanding of ciliary resorption in the past 15 yr. Several kinases that regulate the ciliary axonemal disassembly have been identified (e.g., AurA, PLK1, Nek2, Cdk5) (Pugacheva et al, 2007; Wang et al, 2013; Kim et al, 2015). AurA regulates ciliary resorption by activating HDAC6 and destabilizing the ciliary axoneme (Pugacheva et al, 2007). The basal body–localized PLK1 promotes ciliary disassembly by activating the AurA-HDAC6 pathway and the phosphorylation of kinesin KIF2A (Lee et al, 2012; Wang et al, 2013; Miyamoto et al, 2015). Nek2, which is also localized at the basal body, modulates cilium disassembly through the microtubule depolymerization and phosphorylation of KIF24 (Spalluto et al, 2012; Endicott et al, 2015; Kim et al, 2015). Cdk5 facilitates ciliary resorption by preventing the ciliary microtubule assembly through phosphorylation (and subsequent ubiquitin degradation) of Nde1 at the basal body (Maskey et al, 2015). The current study identifies MAST4 as a novel cilium-localized kinase that promotes ciliary resorption. We used a combination of epistatic functional assay, quantitative fluorescence imaging analysis, Cdc42 activation assay, and transferrin endocytosis assay to comprehend the mechanism that underlies MAST4-regulated ciliary resorption. The results of these studies showed that MAST4 accelerates ciliary resorption by promoting phospho-(T94)

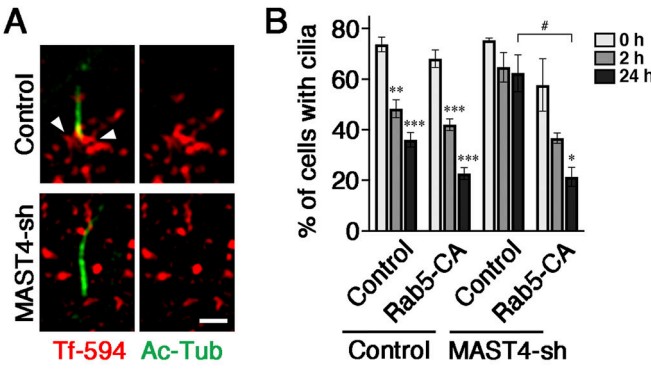

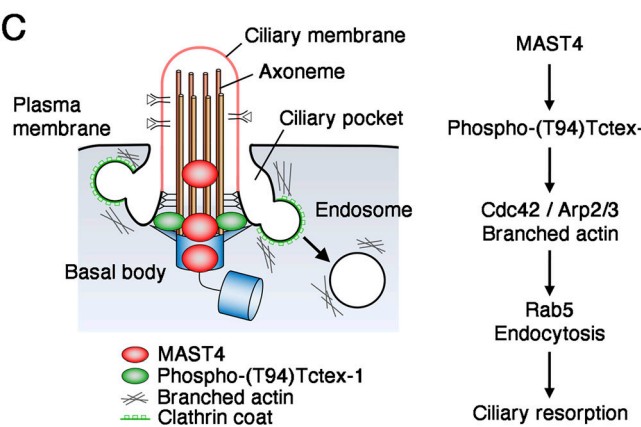

**Figure 6. MAST4 regulates the periciliary membrane endocytosis working model.**

**(A)** Representative SR-SIM images show Ac-Tub–labeled cilium (green) and the periciliary signal of internalized Alexa Fluor 594–conjugated transferrin (Tf-594) (red) in RPE-1 cells transfected with GFP alone (control), and GFP together with MAST4-sh1 1 h post–IGF-1 (10 nM) stimulation. Ac-Tub was detected by Alexa Fluor 405–conjugated secondary antibody and is pseudo-colored in green for presentation. Reprehensive images of 14 cilia (in control) and 8 cilia (in MAST-sh1–transfected cells) are shown. Scale bar: 1 $\mu$m. **(B)** Ciliary resorption assay of RPE-1 cells transfected with GFP alone (control), or MAST4-sh1-IRES-GFP (MAST4-sh1) together with FLAG-tagged Rab5$^{Q79L}$ (Rab5-CA). The histogram shows the percentage of GFP$^+$ cells with Ac-Tub–labeled primary cilia at the indicated time points. Values are the mean ± S.E.M. *$P < 0.05$, **$P < 0.01$, and ***$P < 0.001$, one-way ANOVA followed by Tukey's test versus 0 h for each group. #$P < 0.05$, two-way ANOVA followed by Bonferroni's test. $n = 100$ cells in each experiment, with three independent experiments. **(C)** Schematic diagram shows the periciliary location and the processes by which MAST4 mediates the ciliary resorption, through the activation of phospho-(T94)Tctex-1, Cdc42/Arp2/3 activity (probably branched actin organization), and periciliary membrane endocytosis.

Tctex-1–mediated pathways, including the activation of Cdc42 GTPase and periciliary membrane endocytosis regulated by Rab5.

The existing model suggests three main steps involved in the process of ciliary resorption: (1) the disassembly of the ciliary axonemal microtubule, (2) the prevention of the ciliary (re)assembly, and (3) the elimination of the extra ciliary luminal/membrane components resulting from the axonemal disassembly (Liang et al, 2016; Hsu et al, 2017; Patel & Tsiokas, 2021). Emerging evidence (Clement et al, 2013; Saito et al, 2017) and the present study suggest that periciliary membrane endocytosis actively participates in ciliary resorption, rather than simply removing the unnecessary ciliary membrane components generated during cilium resorption.

The endosome has long been recognized as a signaling hub that couples the signals sensed by the cell surface receptor, the distribution of intracellular signaling components, and the gene regulation in the nucleus (Scita & Di Fiore, 2010). Accordingly, the downstream signaling components of several ciliary membrane receptors are localized in the endosomes at the ciliary base. For example, phospho-SMAD2/3, the notable downstream effector of the TGF-$\beta$ receptor (a ciliary membrane protein), is abundantly accumulated in the periciliary endosomes during ciliary disassembly (Clement et al, 2013; Saito et al, 2017). It remains to be investigated how the internalized surface signals modulate a cascade of cellular events that ultimately harness the process of ciliary resorption.

MAST4 was initially identified as a protein that co-immunoprecipitated with Tctex-1 during the first phase of ciliary resorption (Saito et al, 2017) (Table S1). The present study experimentally validates the physical interaction between MAST4 and Tctex-1 using co-immunoprecipitation of HEK293-transfected proteins and pull-down assays using bacterial extracts containing recombinant proteins. The latter revealed that the kinase domain of MAST4 binds to Tctex-1 with the highest affinity. The site-directed mutation analyses further showed that the R503 and D504 residues in this domain are critical for the ciliary resorption function of MAST4. The expression of MAST4$^{R503A}$ and MAST4$^{D504A}$ phenocopies the MAST4 KD-mediated ciliary resorption blocking and ciliary base emergence of phospho-(T94)Tctex-1. Analogous to the other AGC kinase family members, MAST4$^{R503A}$ and MAST4$^{D504A}$ are likely to be kinase-dead. These results collectively suggest that the R503 and D504 residues in the catalytic motif of MAST4 are pivotal for ciliary resorption through the activation of the phospho-(T94)Tctex-1–mediated pathway.

The above studies tempted us to propose MAST4 is the kinase that phosphorylates Tctex-1. The direct evidence supporting this claim, however, remains lacking because of several technical reasons. First, the currently available antibody is not sensitive enough to detect phospho-(T94)Tctex-1 on immunoblots. Second, probably because of its large size, it is challenging to synthesize a full-length MAST4 protein in *E. coli* to perform the phosphorylation assay using the purified Tctex-1 as a substrate. Also, we cannot rule out the existence of another kinase that phosphorylates Tctex-1 and this might contribute to the basal-level ciliary base signal of phospho-(T94)Tctex-1 in MAST4-KD cells. Phospho-(T94)Tctex-1 is concentrated at the ciliary base, whereas MAST4 is broadly distributed along the cilium. The ciliary location(s) where Tctex-1 is phosphorylated, either by MAST4 or by other yet-to-be-identified kinases, remain(s) unknown.

Ciliary resorption has been closely linked to cell cycle progression, cell fate determination during corticogenesis, and cancer (Sung & Li, 2011; Fabbri et al, 2019; Peixoto et al, 2020; Shiromizu et al, 2020). Phospho-(T94)Tctex-1's ciliary resorption function is causally linked to the $G_1$-S transition during the cell cycle and cell fate during corticogenesis in neural progenitor cells (Li et al, 2011; Sung & Li, 2011; Yeh et al, 2013). Cdc42 and actin dynamics also have reported roles in cell cycle regulation (Olson et al, 1995; Lamarche et al, 1996; Reshetnikova et al, 2000; Villalonga & Ridley, 2006; Heng & Koh, 2010). It is interesting to speculate that MAST4 also has a role

in corticogenesis by activating the phospho-(T94)Tctex-1–mediated pathway.

Loss of primary cilia has been observed in several breast cancer cell lines and primary breast cancer tissues (Yuan et al, 2010). Of interest, both Tctex-1 (Huang et al, 2023) and MAST4 (Beretov et al, 2015) have been proposed to be potential prognosis markers of breast cancer, because of their up-regulated expression in human primary breast cancer tissues. Whether the overexpression of these proteins contributes to the etiology of the disease through disturbing the ciliary dynamics and cell cycle control warrants future investigation.

Several MAST4 substrates have been previously reported. One of them, phosphatidylinositol 3,4,5-trisphosphate 3-phosphatase (PTEN), is involved in ciliary resorption. PTEN negatively regulates ciliary resorption in RPE-1 cells by suppressing the $CK1\varepsilon$-dependent PLK1-AurA pathway (Shnitsar et al, 2015). Other MAST4 substrates including forkhead box transcription factor O1 (FOXO1), Ets-related molecule, and Sox9 have been linked to self-renewal, cell proliferation, and cell differentiation (Valiente et al, 2005; Gongol et al, 2017; Zhang et al, 2018; An et al, 2019; Lee et al, 2021; Kim et al, 2022). The conceivable crosstalk between multiple MAST4 substrates during ciliary resorption and other cellular events is of future interest.

In conclusion, our identification of MAST4 as a novel kinase that promotes ciliary resorption expands our toolbox to manipulate this cellular process and develop strategies for treating diseases caused by dysregulated ciliary dynamics.

# Materials and Methods

## Reagents–cell cultures, chemicals, antibodies, primers, and plasmids

RPE-1 cells were obtained from the American Type Culture Collection (cat. #CRL-400). HEK293 cells were obtained from the Cell Resource Center for Biomedical Research, Tohoku University (Miyagi, Japan). DMEM (#08458-45) was purchased from Nacalai Tesque. FBS was purchased from PAA Laboratories GmbH. DMEM/Ham's F-12 (DMEM/F12) (#048-29785), 100 mM pyruvate (#190-14881), ImmunoStar LD, and antibodies including mouse anti-DYKDDDDK/FLAG antibody (#012-22384), mouse anti-glyceraldehyde-3-phosphate dehydrogenase (GAPDH) antibody (#016-25523), and mouse anti-GFP antibodies (clone mFX73; #012-20461) were purchased from FUJIFILM Wako Pure Chemical. Anti-Ac-Tub (clone 6-11B-1; #T6793) and anti-γ-Tub (clone TUB2.1; #T6557) antibodies were purchased from Sigma-Aldrich. TRIzol (#15596026), Lipofectamine LTX (#15338-100), Alexa Fluor 594–conjugated transferrin (#T13343), Alexa Fluor dye–conjugated secondary antibodies, 4′,6-diamidino-2-phenylindole (DAPI), protein G Sepharose (#17-0886-01), and glutathione Sepharose 4B (#17-0756-01) were from Thermo Fisher Scientific. Rabbit anti-MAST4 antibody (#A302-397A) was purchased from Bethyl Laboratories. Rabbit anti-GFP antibody (#598) was from Molecular and Biological Laboratories. Mouse anti-mCherry antibody (#677702) was from BioLegend. Chicken anti-mCherry antibody (#ab205402) was from Abcam. CF dye–conjugated secondary antibodies were purchased from Biotium. Mouse anti-MBP antibody (clone 8G1; #2396), rabbit anti-Cdc42 antibody (#2462), HRP-conjugated anti-mouse IgG (cat. #7076), and HRP-conjugated anti-rabbit IgG (#7074) were purchased from Cell Signaling Technology. The PrimeScript RT reagent (#RR037A) and TB Green Premix Ex Taq II (#RR820A) were purchased from Takara Bio. Phusion High-Fidelity DNA Polymerase (#M0530S) was purchased from New England Biolabs. The QuikChange Site-Directed Mutagenesis Kit (#200519) was purchased from Stratagene (Agilent Technologies). Rabbit anti-Tctex-1 antiserum (Saito et al, 2021) and rabbit anti-phospho-(T94)Tctex-1 antibody were homemade (Li et al, 2011).

The plasmids encoding ARPC2-shRNA, FLAG-Tctex-1$^{WT}$, FLAG-Tctex-1$^{T94E}$, FLAG-Tctex-1$^{T94A}$, HA-Cdc42$^{T17N}$, HA-Cdc42$^{G12V}$, and FLAG-Rab5$^{Q79L}$ were previously described (Saito et al, 2017). MAST4 shRNA plasmids were made by inserting the annealed shRNA targeting sequences (MAST4-sh1 and MAST4-sh2) behind a U6 promoter in the pCAG-IRES-GFP plasmid. pCAG-MAST4-sh1 (without IRES-GFP) and pCAG plasmids were used in Cdc42 activity assays. The cloning of full-length MAST4$^{WT}$ and site-directed mutants was described below. The sequences of the oligonucleotides used in this study are listed in Table S2.

## qRT–PCR

RNAs of RPE-1 cells were purified using TRIzol. For qRT-PCR, first-strand cDNA was synthesized from total RNAs using an Oligo(dT)$_{20}$ primer and PrimeScript First-Strand Synthesis System (Thermo Fisher Scientific). The PCRs were performed in a 20 $\mu$l reaction mixture using TB Green Premix Ex Taq II Master Mix (Thermo Fisher Scientific). Each reaction was performed in triplicate, and the cycle threshold ($Ct$) values of interesting genes were normalized to a housekeeping gene *GAPDH*. The relative mRNA levels of *MAST1-4* were calculated by the $2^{-\Delta\Delta Ct}$ method. The PCR products were also analyzed by gel electrophoresis to confirm the amplification of a single predominant band of expected sizes.

## Cell transfection

All cell cultures were maintained in a humidified atmosphere at 37°C and 5% $CO_2$. HEK293 cells were grown in DMEM supplemented with 10% heat-inactivated FBS and transfected using Lipofectamine LTX. RPE-1 cells were grown in DMEM/F12 supplemented with 10% heat-inactivated FBS and 1 mM pyruvate. ECM 830 Square Wave Electroporation System (BTX Harvard Apparatus) or Neon Transfection System (Thermo Fisher Scientific) was used to transfect RPE-1 cells.

## Ciliary resorption assay

The cilium assembly and resorption assays were carried out as previously described (Saito et al, 2017, 2018). Briefly, the transfected RPE-1 cells were plated on coverslips for 8–12 h before serum starvation for 36 h to induce ciliogenesis. Afterward, the serum-containing medium was added back to trigger ciliary resorption, and the cells were harvested at the indicated time points. The number of GFP$^+$-transfected cells expressing Ac-Tub–labeled cilium (also see below) was counted under epifluorescent microscopy in a double-blind fashion. More than 100 transfected cells were counted for each condition in three independent experiments.

## Immunostaining, imaging, and immunofluorescence quantification

For immunostaining, cells grown on coverslips were fixed by 4% PFA in PBSc/m (PBS plus 0.2 mM $Ca^{2+}$ and 2 mM $Mg^{2+}$), quenched (50 mM $NH_4Cl$, 10 min), blocked (0.5% BSA, 0.01% Triton X-100, and DAPI in PBSc/m for 30 min), and incubated with primary antibodies for 1 h at room temperature in blocking buffer. After three PBSc/m washes, Alexa Fluor dye– or CF dye–conjugated secondary antibodies were incubated for an additional 45 min at room temperature before mounting. Whenever γ-tubulin was co-labeled, the PFA-fixed cells were further treated with chilled methanol for 30 s at –20°C. To stain phospho-(T94)Tctex-1, RPE-1 cells were first incubated with 0.1% Triton X-100–supplemented PHEM buffer (100 mM PIPES, 100 mM Hepes, 2 mM $MgSO_4$, 0.2 mM EDTA, and 4 mM EGTA, pH 6.9) for 30 s at room temperature before 4% PFA fixation for 10 min.

Confocal microscopic images were acquired using a Zeiss LSM780 confocal microscope by a 63× objective lens with a z-interval of 0.25 $\mu m$. Super-resolution Airyscan images were acquired using a Zeiss LSM800 confocal microscope and by a 63x objective lens with a z-interval of 0.15 $\mu m$.

For quantitative immunofluorescence analysis, the confocal fluorescence intensity of MAST4-mCherry across the entire cilium and ciliary base was plotted using Zen software (Zeiss). The representative line scans from the images of 10–16 cilia are shown. To quantify the ciliary basal signal of phospho-(T94)Tctex-1, the total fluorescence of the region of interest, that is, the proximal end of Ac-Tub–labeled cilium, was quantified in the projection confocal images of 1.25-$\mu m$-thick Z-series sections of cilium-containing regions ($x = 1\ \mu m$, $y = 1\ \mu m$) in randomly selected cells using NIH ImageJ 64 software. The numbers of analyzed cilia are indicated in the figure legend. All experiments were repeated at least three times. The ciliary length was measured using the acquired confocal Ac-Tub images on NIH ImageJ 64 software. For presentation purposes, the brightness and the background of the images were adjusted by Photoshop Elements 12 software (Adobe Systems).

## Molecular cloning of MAST4 and site-directed mutagenesis

We used Phusion High-Fidelity DNA Polymerase to amplify the N-terminal (MAST4-N) and C-terminal (MAST4-C) halves of human MAST4 cDNA using the cDNAs of HEK293 cells as a template (primers are listed in Table S2). These fragments were then sequentially subcloned into a pCAGIG vector to obtain the plasmid encoding full-length MAST4[WT] followed by IRES-GFP, that is, pCAGIG-MAST4[WT]. QuikChange Site-Directed Mutagenesis was carried out in the pCAGIG-MAST4[WT] plasmid using the primers listed in Table S2. To generate the pCAGIG-MAST4-mCherry plasmid, the mCherry-coding sequence was added to the 3' of the MAST4[WT] cDNA using a standard molecular cloning technique.

## Transferrin uptake assay

Transferrin uptake assays were performed as previously described (Saito et al, 2017). In brief, serum-starved RPE-1 cells were treated with 10 nM IGF-1 and 25 $\mu g$/ml Alexa Fluor 594–conjugated transferrin in 0.5% BSA-containing DMEM at 37°C for 1 h. The cells were then placed on ice, washed twice with cold PBSc/m, incubated with mild acid buffer (0.5% acetic acid and 500 mM NaCl, pH 5.0) for 2 min, and then washed again with cold PBSc/m twice before 4% PFA fixation. Previous studies showed that this acid wash protocol removed >90% of cell surface–bound transferrin (Johnson et al, 2001). The fixed cells were immunolabeled for Ac-Tub as described above and followed by acquiring images using a super-resolution structured illumination microscope (SR-SIM) (N-SIM; Nikon) with a Nikon SR Apo TIRF 100x/1.49 oil objective equipped with an iXon3 EMCCD camera (DU-897E; Andor Technology) on a Nikon N-SIM system with excitation laser light of 405- and 561-nm wavelengths. Images were reconstructed using NIS-Elements AR software (Nikon).

## Co-immunoprecipitation and immunoblotting assays

Sample preparation for the co-immunoprecipitation assay was performed with a brief modification from our previous report (Saito et al, 2019). Transfected HEK293 cells expressing MAST4[WT]-mCherry and FLAG-Tctex-1[WT] were rinsed twice with cold STE buffer (50 mM Tris–HCl, 150 mM NaCl, and 2 mM EDTA, pH 7.4) and then lysed in STET buffer (STE buffer supplemented with 1% Triton X-100, protease inhibitor cocktail, 1 mM phenylmethylsulfonyl fluoride, and phosphatase inhibitor cocktail) at 4°C for 30 min. The cell lysates were centrifuged (15,000$g$, 10 min, 4°C), and the post-nuclear supernatants were pre-cleared using protein G Sepharose (30 min, 4°C). The cleared supernatants containing equal amounts of proteins were incubated with 1 $\mu g$ of mouse anti-mCherry antibody (or mouse anti-GFP antibody for control) overnight at 4°C. Pre-cleared protein G Sepharose was then added for an additional 1 h at 4°C. Afterward, protein G Sepharose was washed three times with cold STET buffer and once with STE buffer, and eluted with Laemmli sample buffer (62.5 mM Tris–HCl, 2% SDS, 5% 2-mercaptoethanol, 10% glycerol, and 0.5 mg/ml bromophenol blue, pH 6.8). The eluted proteins were analyzed by electrophoresis followed by immunoblotting assays using a standard method. Briefly, the protein blots were incubated with primary antibodies overnight at 4°C followed by HRP-conjugated secondary antibodies for 1 h at room temperature and the ImmunoStar LD (peroxidase luminescent substrate). Chemiluminescent signals were digitally recorded using ChemiDoc MP (Bio-Rad). The intensity of the bands was quantified using Image Laboratory software (Bio-Rad).

## Recombinant protein and pull-down assay

*E. coli* BL21 cells expressing MBP-Tctex-1 or GST-MAST4 fragments were homogenized using French Pressure Cell Press (FA-078; SLM Aminco) and centrifuged at 15,000$g$ for 15 min at 4°C. The supernatants were analyzed on SDS–PAGE and stained with Coomassie brilliant blue. The yield of the recombinant proteins was determined based on a serially diluted known concentration of BSA run on the same gel and quantified by Odyssey Infrared Imager (LI-COR, NE). For the pull-down assay, the supernatants containing 10 $\mu g$ of MBP-Tctex-1 were mixed with the supernatants containing 5 $\mu g$ of the GST-MAST4 fragment in buffer A (1x TBS supplemented with 0.1% NP-40, 0.2 mg/ml BSA, and 1 mM dithiothreitol) at a final volume of 300 $\mu l$. The mixtures were rotated for 2 h at 4°C and then incubated

with glutathione Sepharose 4B beads for 1 h at 4°C. After that, the beads were washed in buffer A thrice and in TBS once, and finally eluted using Laemmli sample buffer. A fraction of the eluates was subjected to electrophoresis. The gels were stained with Coomassie brilliant blue to verify the input of GST-MAST4 fragments by Odyssey Infrared Imager. Another fraction of the eluates was subjected to MBP immunoblotting assays.

### Cdc42 activity assay

Transfected RPE-1 cells were lysed in the lysis buffer (50 mM Tris–HCl, 10 mM $MgCl_2$, 300 mM NaCl, and 2% NP-40, pH 7.5, plus protease inhibitors and phosphatase inhibitors). The post-nuclear supernatants (13,000$g$, 15 min, 4°C) containing 40 $\mu$g of total proteins were subjected to the Cdc42 activity assays according to the manufacturer's manual (Cdc42 Activation Assay Biochem Kit; Cytoskeleton, cat. #BK034-S). Briefly, to pull down the active form of Cdc42, cell lysates were incubated with glutathione beads conjugated with a GST fusion of the PBD region of PAK (1 h, 4°C). The beads were washed with lysis buffer once and with wash buffer (25 mM Tris–HCl, 30 mM $MgCl_2$, and 40 mM NaCl, pH 7.5) once, and then eluted in 1x Laemmli sample buffer. One-third of the elutes and total cell lysates (containing 20 $\mu$g of total proteins) were electrophoresed on the same gels, transferred, and probed with anti-Cdc42 antibody. The Cdc42 signals on the immunoblots were detected by ChemiDoc MP and quantified by Image Laboratory software. The ratios of activated Cdc42 to total Cdc42 in RPE-1 cells harvested at different time points were compared considering the 0-h time point as 100.

### Protein structure prediction

The structure of the MAST4 kinase domain was predicted using the AlphaFold2 tool (https://colab.research.google.com/github/sokrypton/ColabFold/blob/main/AlphaFold2.ipynb#scrollTo=kObIAo-xetgx). The predicted structure was displayed with and without side chain of amino acids.

### Statistics

Prism software (ver8.4, GraphPad Software) was used for graph drawing and statistical analysis. Data are presented as the mean ± S.E.M. from at least three representative independent experiments. One-way analysis of variance (ANOVA) followed by Tukey's test (as a *post hoc* test), two-way ANOVA followed by Bonferroni's test (as a *post hoc* test), or $t$ test was performed. Statistical significance was set at $P < 0.05$. The numbers of samples are indicated in the figure legends.

## Supplementary Information

## Acknowledgements

This research was funded by Grants-in-Aid for Scientific Research from the Japan Society for Promotion of Science (Nos. 18K06125 and 21K06059 to M Saito; No. 21K05427 to G-i Atsumi) and the Takeda Science Foundation (to M Saito), and NIH RO1 EY032966, EY029429, and the Research to Prevent Blindness (to C-H Sung). We would also like to acknowledge Dr. Teruyuki Yanagisawa (Tohoku University Graduate School of Medicine) for providing guidance and feedback throughout this project and Mrs. Mitsuyo Suzuki, Mrs. Sayuri Saito, and Mr. Akihiko Ichikawa (Carl Zeiss) for their technical support.

## Author Contributions

K Sakaji: resources, data curation, formal analysis, and investigation.
S Ebrahimiazar: resources, data curation, formal analysis, investigation, and writing—original draft.
Y Harigae: resources, data curation, formal analysis, and investigation.
K Ishibashi: data curation, formal analysis, and investigation.
T Sato: conceptualization and supervision.
T Yoshikawa: conceptualization and supervision.
G-i Atsumi: conceptualization, supervision, and funding acquisition.
C-H Sung: resources, data curation, supervision, funding acquisition, project administration, and writing—original draft, review, and editing.
M Saito: conceptualization, resources, data curation, software, formal analysis, supervision, funding acquisition, validation, investigation, visualization, methodology, project administration, and writing—original draft, review, and editing.

### Conflict of Interest Statement

The authors declare that they have no conflict of interest.

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
