## [Reviewer comments · Life Science Alliance]

Life Science Alliance

MAST4 promotes primary ciliary resorption through phosphorylation of Tctex-1

Kensuke Sakaji, Sara Ebrahimiazar, Yasuhiro Harigae, Ken-ichi Ishibashi, Takeya Sato, Takeo Yoshikawa, Gen-ichi Atsumi, Ching-Hwa Sung, and Masaki Saito

DOI: <https://doi.org/10.26508/lsa.202301947>

Corresponding author(s): Masaki Saito, Teikyo University and Ching-Hwa Sung, Weill Medical College of Cornell University

Review Timeline:

Submission Date:	2023-01-24
Editorial Decision:	2023-03-09
Revision Received:	2023-08-07
Editorial Decision:	2023-08-23
Revision Received:	2023-08-27
Accepted:	2023-08-28

Transaction Report:

March 9, 2023

Re: Life Science Alliance manuscript #LSA-2023-01947-T

Dr. Masaki Saito
Teikyo University
Faculty of Pharma-Sciences, Department of Molecular Physiology and Pathology
2-11-1 Kaga
Itabashi-ku, Tokyo 173-8605
Japan

Dear Dr. Saito,

Thank you for submitting your manuscript entitled "MAST4 promotes primary ciliary resorption by phosphorylating Tctex-1" to Life Science Alliance. The manuscript was assessed by expert reviewers, whose comments are appended to this letter. We invite you to submit a revised manuscript addressing the Reviewer comments.

Thank you for this interesting contribution to Life Science Alliance. We are looking forward to receiving your revised manuscript.

Sincerely,

B. MANUSCRIPT ORGANIZATION AND FORMATTING:

Reviewer #1 (Comments to the Authors (Required)):

1) This manuscript describes the identification of ciliary MAST4 and its role in ciliary resorption. MAST4 KD cells resorb cilia with delayed kinetics, and MAST4 overexpression accelerates population-level ciliary disassembly. This paper presents data indicating that MAST4 physically interacts with Tctex-1 and identifies the responsible protein fragment. Sakaji et al hypothesize that MAST4 phosphorylates Tctex-1, with supporting data that in MAST4 KD cells phosphomimetic Tctex-1 restores near control level population resorption kinetics. A selection of point mutants reveals the amino acids linked to defects in resorption. In the absence of MAST4, Cdc42 activation is perturbed, providing some rationale on how MAST4 fits into the larger known mechanisms underlying ciliary resorption.

Overall the manuscript was well written, provides new information about ciliary resorption, and will be interest to Life Science Alliance readers. All comments offered herein are in good spirit to both help the authors and future readers reproduce these experiments.

2) Main points:

- MAST4 accelerates ciliary resorption

Yes, the data is supportive. It would be enhanced with better MAST4-mC images as those presented in Figure 1 E-F have much background making it difficult to distinguish signal from noise (especially in E).

- The kinase domain of MAST4 directly binds to Tctex-1

Blots in Fig 2 B are not sufficient to claim endogenous MAST4 co-immunoprecipitates with Tctex-1.

- MAST4 enhances the ciliary transition zone expression of phospho-(T94)Tctex-1

Without details on how these experiments were performed, I cannot evaluate if the data support the claims or not.

- The kinase activity of MAST4 is required for Tctex-1-mediated ciliary resorption

The data supports that the kinase activity of MAST4 is required for ciliary resorption. Again, need details on qIF to interpret properly.

- MAST4 regulates ciliary resorption by activating the downstream effectors of phospho-(T94)Tctex-1

The data supports indirect activation of downstream effectors & upstream involvement of MAST4 with Tctex-1. I cannot interpret how the GFP-F experiment was quantified due to lack of details.

3) Major points: For interpretation & reproducibility, this manuscript needs a M&M section on quantitative immunofluorescence for figures 3,4,6. Presumably, for Fig3-4, you've measured the intensity of the signal in the channel of interest? This should really be done with a reference channel, i.e., a known TZ marker to make an ROI for measuring your channel of interest. Did you subtract background fluorescence? How did you take these measurements with puncta of variable size? Figure 6C-D is super unclear how one quantifies such "periciliary" signal. We need clear methods on how this data was generated.

Figures 3 & 4: There's a puncta for P-(T94)Tctex-1 in the KD & point mutants so presumably some other kinase is phosphorylating this substrate. Please add to the discussion that MAST4 is not solely responsible (unless you have additional supportive data to include) & anything that's known about other kinases.

Additional specific issues:

General: some wording throughout like "expression of X protein at ciliary TZ"-I think you mean localization? Surely not implying gene expression, right? I'm also unclear on how a transient interaction like that of a kinase & substrate can be stable enough for a pulldown. Some discussion might help here.

Figure 1

B: It doesn't say here, but I'm assuming these measurements were made off ac-tub signal. That's a bit problematic since deacetylation happens before resorption as the authors cite in Pugacheva et al.

E: That's a lot of background for MAST4-mC. The fluorescence intensity line plot should be segmented to cover the length of the

cilium as defined by ac-tub staining. As displayed, it looks like mC signal ends at tip, but as drawn, it looks like it's due to the straight line. If not, fix the line in the first panel to reflect appropriately. Honestly, not sure if this adds much though will all the background. What's the true signal here? Would like to see larger whole cell pic to get a better idea of signal-to-noise.

F: need scale bar & asterisks definition

Figure 2

A & B: These blots leave much to be desired. With signals like these, this reviewer requests full panel of these membranes. Tctex-1 "signal" is especially problematic as it doesn't even really look like a band, but rather a smudge from signal above. Shouldn't these weights be different for MAST4-mC & endogenous MAST4? Looks nearly identical.

D: Signal for frag 1 too in anti-MBP blot? Why aren't we seeing a clear 56 kDa band in lane 2 indicative of MBP-Tctex-1?

Figure 3

A: for consistency, please label the figure with MAST4-sh1

B: missing scale bar

Figure 4

C: Of all the puncta in these little panels, you're quantifying those that overlap with ac-tub signal?

Figure 5

C-F: Need a better explanation of how you're measuring active Cdc42 (I had to google it but it should be in the paper. If you're including densitometry of blots & running stats, this reviewer wants to see full panels in the supplement of all blots used to generate D & F. It's also unclear how you're measuring relative levels: is the active & total Cdc42 on the same membrane? I should hope so. Need more in M & M than Ins 466-467.

Figure 6

B: How are you differentiating/identifying internalized Tf-594. From this data, I cannot believe those specific puncta are internalized vs all the others.

C: This experiment seems shaky from what's presented. How are you claiming this signal is from the ciliary pocket?? See major points above regarding quantification of this experiment.

D: Figure legend says this is in %-please indicate likewise on the figure

EV2 Why isn't there an endogenous MAST4 signal in B while you can see it in A? GAPDH loading is similar so one should see it.

EV3 What's going on with K506A signal here? Need to show clearly that it is expressed for other experiments.

EV4 Way too much mC background here to say where this signal is coming from! One could generate line intensity plots anywhere in these small panels. This is not acceptable data to support subcellular localization. The figure isn't labeled with mutants. No scale bar definition in figure legend.

Ln 111: These are not isoforms of one protein but rather a family of kinases as you said in the intro. This sentence leads one to believe these are isoforms of MAST (even a thing)? Consider "MAST4 is the predominant kinase of the MAST family"

Ln 140: is this ciliary pocket staining? I'd like to see a secondary pocket marker to be convinced

Ln 140-142: how do these data support that MAST4 plays an imp't role in only the early phasic resorption 0-2hr when resorption is also perturbed at 24 hours?

Ln 144: seems like an illogical jump here if you read results first. Consider adding a similar statement about the interaction identified in a previous paper here

Ln 175: Haven't shown MAST4 phosphorylates Tctex-1-either provide direct data or adjust to "likely phosphorylates" or the like

Ln 181: AGC? Define

Ln 201-202: cannot say from these images that anything moved from basal body to axoneme

Ln 203: expression→ localization? "Variants resulted in reduced signal from"

Ln 442→ TIRF?

Ln 704: scale bar in F also 1?

Ln 724-725: could put kDa in the figure easier for the reader to interpret the blot below- to scale with C

Ln 776: PAK-PBD beads? Please explain these beads so readers don't have to google to figure out your experiments.

Ln 812: major member? Consider "predominately expressed"

Reviewer #2 (Comments to the Authors (Required)):

MAST4 promotes primary ciliary resorption by phosphorylating Tctex-1
K.Sakaji., S. Ebrahimiazar, Y. Harigae, T. Sato, T. Yoshikawa, C. Sung, M. Saito

1. A short summary of the paper, including description of the advance offered to the field.

In this work Sakaji et al. study the phenomenon of primary cilia resorption in a cell line in culture. In particular, they address the function of MAST4 (a Ser/Thr kinase) in the process, paying special attention to its interaction with Tctex-1 as a potential substrate. Tctex-1 is a protein that was previously identified as a participant in cilia resorption. The phosphorylation of Thr94 in Tctex-1 (T94) is necessary for the activation of downstream effectors involved in ciliary resorption. In this work the authors determine that MAST4 is involved in ciliary resorption, interacts with Tctex-1 by its kinase domain and that the downstream mechanisms involved in ciliary resorption are the ones triggered by Tctex-1 phosphorylation, thus proposing Tctex-1 a substrate of MAST4. The data obtained in this work provides new information about ciliary resorption, a process that is still not completely understood. It is clear from their work that MAST4 is another player in ciliary resorption, and although they provide evidence of interaction between this kinase and Tctex-1, I think some other simple experiments would further support the claim that MAST4 activity in ciliary resorption is due to Tctex-1 phosphorylation.

2. For each main point of the paper, please indicate if the data are strongly supportive. If not, explicitly state the additional experiments essential to support the claims made and the time frame that these would require.

a) MAST4 accelerates ciliary resorption

The participation of MAST4 in ciliary resorption is clearly supported by knockdown (KD) and overexpression (OE) experiments and the evaluation of the cilia length and % of ciliated cells in transfected cell lines. The conclusion that MAST4 is localized at the ciliary pocket needs to be further confirmed using a marker of the ciliary pocket in the immunofluorescence (IF). This marker was used in Figure 6 and thus should be included in the IF.

b) The kinase domain of MAST4 directly binds to Tctex-1

The authors observed the interaction between MAST4 and Tctex-1 when they overexpressed both proteins, and also with the endogenous proteins. The WB showing the immunoprecipitation of endogenous proteins, which reflects physiological interaction, is not of very good quality, it would be great if it could be improved. Moreover, the authors mentioned in the introduction of the present manuscript that the interaction of MAST4 and Tctex-1 was observed in a previous paper (Saito et al. 2017), in the proteomic analysis of material immunoprecipitated with an anti-Tctex-1 antibody. I could not find this result in the cited reference. It would be nice, if possible, that the authors could include the proteomic data (the corresponding author of the present manuscript is the first author of the one involving proteomics), as another evidence of the interaction between the two proteins. The evidence that the kinase domain of MAST4 is the one involved in interacting with Tctex-1 comes from pull-down experiments where Tctex-1 and different fragments of MAST4 were expressed in bacteria. The results are convincing. However, there is a contradiction between what is described in Materials and Methods and in Legend of Figure 2D. Whereas in M&M it is said that crude extracts from bacteria expressing MAST-fragments was incubated with extracts from bacteria expressing Tctex-1, in Legend of Figure 2D says that purified MAST4 fragments were incubated with purified Tctex-1. It is important to clarify which of the two procedures was used. In case the authors have used crude extracts, they cannot assure that the interaction between the MAST4 kinase domain and Tctex-1 is direct. Please check. I think Figure 2D could improve very much if the pull down of the different MAST4 fragments is visualized by Western Blotting with an anti-GST antibody instead of Coomassie staining.

c) MAST4 enhances the ciliary transition zone expression of phospho-(T94) Tctex-1

First of all, I do not like the use of the word "expression" when referring to localization of a protein. Expression maybe confused with translation, that does not occur at the cilium. Please change "expression" for localization.

The effect of MAST4 KD is reversed by the overexpression of Tctex-1T94E but not the WT or the Tctex-1T94A proteins, suggesting that Tctex-1 could be a substrate of the kinase. I think the results support the idea but other evidences are needed in order to establish that MAST4 phosphorylates Tctex-1. As the authors mentioned in the Discussion section, MAST4 is a big protein, difficult to produce recombinantly so as to test its activity on Tctex-1 directly. However, as this point is very relevant for the work, I suggest that an effort has to be made in order to have more evidence of MAST4 phosphorylating Tctex-1. I am proposing this because there are at least two experiments, that look relatively simple, that could shed light on this point:

i) it would be very informative to see what happens with Tctex-1 phosphorylation levels in MAST4 KD cells. This could be studied by WB using the anti-phospho-Tctex-1 antibody as previously done (Saito et al, 2017). The results could further support the idea that MAST4 phosphorylates Tctex-1.

ii) Does Tctex-1T94A suppress the acceleration of ciliary resorption caused by overexpression of MAST4?

MAST4 KD also decrease the serum-induced localization of phospho-Tctex-1 at the transition zone (TZ). This is clearly supported by the IF images.

There is an issue regarding Tctex-1 localization that is not clear. I did not find in the previous papers (Li et al 2011 and Saito et

al 2017) which is the localization of un-phosphorylated Tctex-1. Is it also present at the TZ and then phosphorylated at this place? Or is it present somewhere else and moved to the TZ upon phosphorylation? Would you be able to answer these questions?

d) The kinase activity of MAST4 is required for Tctex-1-mediated ciliary resorption.

The results supporting that the catalytic domain of the kinase is important for ciliary resorption and phospho-Tctex-1 TZ localization are convincing.

e) MAST4 regulates ciliary resorption by activating the downstream effectors of phospho-(T94) Tctex-1

I consider that this conclusion is well supported by the experiments and the results. However, I found an inconsistency with the previous paper (Saito et al 2017). In Figure 6B there is no effect of the overexpression of the Ccdc42 CA protein, while it was expected to accelerate ciliary resorption. Please comment on that.

Once the authors showed that MAST4 mediates Cdc42 activation and Arp2/3 activity, the rest of the experiments involving endocytosis of the ciliary pocket membrane are a bit redundant, as the link between Cdc42-Arp2/3-endocytosis of the ciliary pocket membrane and ciliary resorption was previously established (Saito et al 2017).

f) Discussion

Line 259-262: Our mechanical studies showed that MAST4 regulates ciliary resorption by promoting the expression localization of phospho-(T94)Tctex-1 in the ciliary transition zone and its downstream activation of Cdc42-ARPC2-mediated ciliary pocket membrane endocytosis.

"Endosomes from the ciliary pocket membrane do not simply remove the ciliary components (e.g., depolymerized tubulin, ciliary membrane) after axonemal microtubule disassembly; they would actively prime ciliary resorption". I do not understand how depolymerized tubulin, a soluble and intracellular protein would be eliminated by endocytosis.

"This idea is supported by the fact that phospho-Smad2/3 accumulates on the endosome membrane, which is generated from the ciliary pocket membrane (Saito et al., 2017)". Please expand on the link between activation of Smad signaling and ciliary resorption as it is not clear from the reference.

"MAST4 certainly controls corticogenesis by phosphorylating (T94)Tctex-1 in the cells". This idea has not been tested in this work, so it should be expressed in a conditional way.

3. Lastly, indicate any additional issues you feel should be addressed (text changes, data presentation, statistics etc.)

Pg 2 line36: change expression for localization

Pg 3, line 52: what is meant by "transition hub"? "Transduction hub" maybe?

Pg 3, line 52: change "cilium-expressing" for ciliary proteins.

Pg3, line 55: change "membranes" for membrane.

Pg 4, line 73: "as well" is redundant with "also" in previous line.

Pg 4, line 75: replace "ciliary" by cilium.

Pg 4, line 79: replace "act on" by act in.

Pg 4, line 84: specify ciliary pocket membrane components.

Pg 4, line 85: add reference after "ciliary resorption"

Pg 5, line 98: re write "isolated as a component in the proteomics of Tctex-1..." as identified an interactor of Tctex-1 in...

Pg 5, line 106: change expression for localization.

Pg 6, line 114: change "different sequences" for different MAST4 sequences.

Pg 6, line 115: the reference Li et al 2011 should be after the word "plasmids" in the previous line.

Pg 6, line 117: move "by immunoblotting" after " We confirmed" in line 115.

Pg 6, lines 119-121: re-write "The length and number of the green fluorescent protein (GFP)+ transfected cells with cilia, labeled with anti-acetylated α -tubulin (Ac-Tub) antibody, were analyzed" as Cilia length and percentage of ciliated cells was determined in GFP positive, transfected cells, using immunofluorescence and confocal microscopy.

Pg 6, line 125: eliminate "the"

Pg 6, line 132: The last sentence does not correspond to what is shown in the figure. % of ciliated cells rather than cilia length is shown in Figure 1D. Please correct.

Pg 7, line 134: replace "ciliary expression" by subcellular localization.

Pg 7, line 139: again, change "expression" for localization the first time and presence the second time.

Pg 7, line 141: again, change "expression" for localization or presence

Pg 8, line162: change "expression" for localization.

Pg 8, line 172: replace "depleted" by decreased.

Pg 8, line 173: eliminate "s" from "zones"

Pg 8, line 175: change "expression" for localization
Pg 11, line 231: change "extracellularly fed" by extracellular

Materials and Methods

Ciliary resorption assay, GFP-F imaging assay, and immunostaining
Please provide details of the temperature of incubation of the antibodies.
Co-immunoprecipitation assay and western blotting
Please provide details of the time and temperature of incubation of the antibodies

Pg 30, line 707-708: Please re-write the following sentence "A representative immunoblot of HEK293 cells overexpressing MAST4WT-mCherry and FLAG-Tctex-1WT was subjected to a co-immunoprecipitation assay" as A representative immunoblot of an immunoprecipitation assay using HEK293 cells overexpressing MAST4WT-mCherry and FLAG-Tctex-1WT.

Pg 30, line 712: idem

Pg 30, line C: Diagram showing MAST4 domains

Pg 31, line 738: Values (C)...is the same for Values in A? Please include.

Legend of Figure 5. "The histogram shows the percentage of GFP+ cells expressing Ac-Tub-labeled primary cilia at the indicated time points". Did you count GFP+ cells or GFP+-HA+ cells?

The same question applies for Figure 6, is GFP+ cells or GFP+-FLAG+ cells?

Legend of Figure 6C: please specify the type of microscopy used (confocal, airyscan, SR-SIM)

Reviewer #3 (Comments to the Authors (Required)):

Reviewer summary

This study follows up on previous findings that Tctex-1 in its T94 phosphorylated state and dynein independent function- regulates cilia resorption by activating cdc42 and regulating the branching dynamics of actin that lead to ciliary resorption. In order to further study the mechanisms underlying Tctex-1 function, this study builds upon the identification of MAST4 as a ligand for Tctex-1 using a proteomics approach. In this work, the authors analyze MAST4 role in ciliary dynamics as well as the physical and functional association between Tctex-1 and MAST4. The data provided support the conclusions of a physical and functional interactions. The study also aims to study the involvement of MAST4 kinase activity and does succeed in providing evidence that MAST4 kinase domains residues are important for Tctex-1 function and Tctex-1 effectors. However more evidence is needed to demonstrate that MAST4 kinase activity is involved in these processes.

Overall, the work is relevant to the field of cilia dynamics, ciliopathies and contributes original information to the understanding of primary cilium regulatory mechanisms. The manuscript is well written and is clear except for the parts where I asked for clarification (mostly in methods description)

The length of the paper is appropriate. Data and literature are succinctly but sufficiently discussed.

The major claims made by this report are that:

- 1.MAST4 is expressed in cilia
- 2.MAST4 and Tctex-1 interact physically and functionally
- 3.The catalytic loop of MAST4 is important for Tctex-1 phosphorylation
- 4.Phosphorylated TcTex-1 is localized to the transition zone of the cilium during resorption
- 5.MAST4 Kinase activity is required for Tctex-mediated cilia resorption

Claims 1-3 are sufficiently supported by the data in this manuscript.

For claims 4 and 5 additional experiments or rephrasing would be necessary to accurately reflect the conclusions allowed by the data.

The statement referred to the kinase activity of MAST4 being required for the observed that needs to be consistent throughout the manuscript. My major concern is related to the section 4 of results (the kinase activity of MAST4 is required for Tctex-1-mediated ciliary resorption). In order to fully support this statement, some direct readout for the kinase activity of MAST4 Wt and the point mutations should be provided, since a phosphoTctex1 antibody is available and is actually used in microscopy in this work a western blotting approach would add evidence, provided that the antibody works in western blot. The phrasing that describes the data currently available seems more accurate both in the figure legend and in the abstract.

Similarly, the conclusion about the localization of pTctex-1, should reflect that it is somewhat speculative (although reasonable) but colocalization with a transition zone markers would strengthen this conclusion.

Reviewer comments:

Methods

Cdc42 activity assay. Please provide more details not referring to previous paper. At least a rationale basis for the assay would be appreciated.

Statistics in Material and methods and figure legends: It has been discussed that presenting data with SEM does not add much meaningful information to the graphical representation. Rather, providing a visual representation of 95% Confidence interval- in addition to the statistical significance test results (p values)- would be more informative

RT-PCR: I am not sure about why "cDNA was PCR amplified with specific primers" and was this followed by real time PCR? How do you keep this quantitative? Perhaps this is 2 different experiments? This is somewhat confusing and should be made more clear.

Results

1. MAST4 accelerates ciliary resorption

Minor comment on Fig 1A-C: control cell resorption in biphasic manner. What does biphasic mean in this context? Could this be explained or rephrased? I am not sure that the dynamics seen Both in A and B is biphasic. It looks more linear. The rest of these panels does support the description in the text.

Line 132; the sentence "These cilia underwent further shortly after 1 h up to 24 h time points" is perhaps mistyped. A word seems to be missing after "further"

Fig 1E. Since there is not any good antibody available to test endogenous distribution of MAST4, a different MAST4 fusion protein with a different tag could be used to verify the observed localization. Also, are these cells stably expressing MAST4-mCherry or are they transiently transfected?

Fig 1F. Asterisks. If the picture shows a single cilium then, the bottom asterisks seem to point to the ciliary pocket but it is unclear what is signaled by the top asterisk

Overall data do support the conclusion that MAST4 plays a role in ciliary resorption. The main weakness lies in the interpretation of the localization data.

2. The kinase domain of MAST4 directly binds to Tctex-1

Fig 2A and 2B suggestion: adding IP mCherry and IP MAST4 labels to the figures would help the reading.

Fig 2D. Possible discussion point: the faint band detected for the pull down of MBP-Tctex-1 with Fr.1 might suggest that other regions of the protein contribute to the interaction.

3. MAST4 enhances the ciliary transition zone expression of phospho-(T94)Tctex-1

The title of this section is a more accurate description of the conclusion supported by the data than the title of Figure 3 legend.

Fig 3A strong data of the functional association and the proposal of MAST4 upstream of phospho-Tctex1

Fig 3B and C. Claiming that phosphoTctex-1 localizes at the transition zone might require a marker for the transition zone (perhaps CEP290?). Otherwise the description of the data would be more accurately described by analyzing Tctex-1 localization at the pericentriolar region or cilia base. Some of these concerns are somewhat addressed in figure EV4. A similar approach could be performed for MAST4WT-mCherry. Also, these data do not fully support the conclusion that MAST4 phosphorylates Tctex1 as stated on Figure 3 legend. The title of the section is more accurate.

4. The kinase activity of MAST4 is required for Tctex-1-mediated ciliary resorption

The prediction of the catalytic motifs of MAST4 kinase domain is based on data about MAST2 kinase domain and it is correct in my opinion. Since alphaFold2 tool is available it would be interesting to see how these predictions map on a predicted structure of MAST4. This would probably have only a visualization value since the functional data shown in 4A is sufficient for the purpose of suggesting that the catalytic activity of MAST4 kinase domain is required for promoting the resorption of cilia.

Is there any other known substrate for MAST4 kinase activity that could be used to corroborate that the mutants have a different activity? This could be performed by a western blot of transfected cells using a phospho-specific antibody (eg phosphoTctex-1, phosphoERM or overall levels of p-Ser) Otherwise since no direct evidence for kinase activity is shown the title of this section and the conclusion might need to reflect that the mutated residues are required for ciliary resorption, rather than the kinase activity.

5. MAST4 regulates ciliary resorption by activating the downstream effectors of phospho (T94)Tctex-1

Fig 5A supports the claims in the text.

An explanation of the cdc42 activity assay would be helpful for the interpretation of this article as a stand-alone set of results.

Fig 6 supports the conclusion that MAST4 acts upstream of known effectors of Tctex-1

Discussion

Line 260. Mechanical might not be the best word here

Line 263 where it says were it should be was

Line 284. Explanation for lack of direct evidence for phosphorylation of Tctex-1 by MAST4. However some experiment could be attempted. Precisely a list of potential readouts for MAST4 kinase activity is shown next.

Line 303. "MAST4 certainly controls corticogenesis by phosphorylating (T94)Tctex-1 in the cells" No reference is given so this is a speculation, albeit reasonable, based on data from this paper so it should be stated as such.

The conclusions are supported partially by data. MAST4 is a regulator of cilia dynamics is supported by the data in this study. However, there is only indirect evidence showing that MAST4 kinase activity is required and that MAST4 phosphorylates Tctex-1. Since there is an anti-phosphoTctex antibody available that was used in IF, it might be possible to use it in a western blot testing the levels of pTctex-1 in the presence of each of the point mutations.

Issues to be addressed:

I explained the details in previous sections but I summarize main concerns here:

Please provide direct evidence of kinase activity of MAST4 wild type and mutants preferably on Tctex-1 or on any of its known substrates. Alternatively, adjust text (mostly title and title section 4 of results and figure 3 title legend) to reflect the data available. The conclusions are more accurately described in the title of the legend for figure 4 and result section 3.

Please provide evidence of colocalization of Phospho-Tctex-1 with a marker for the transition zone of the cilium. Alternatively, adjust text to reflect the data available.

Referee cross comments:

In addition to my previous comments, I agree with the important points raised by both the other reviewers about changing the text of section 3 of results using Localization instead of expression

Point-by-point Responses

Original figure/panel	Revised figure/panel	Issue to be addressed	Reviewer(s)
Figure 1E	Figure 1E	Background signal, low-power view of an entire cell	Reviewer 1
Figure 1F	Figure EV3	Background signal, low-power view of an entire cell	Reviewer 1
Figure 2A	Figure 2A	Full immunoblots are desired	Reviewer 1
Figure 2B	Removed	Signal of the pulled down endogenous Tctex-1 was unclear	Reviewer 1, 2, 3
Fig. 2D	Fig. 2C	Clarity of figure labeling	Reviewer 1, 2
Figure 4C	Figure 4C	Background signal	Reviewer 1
Figure 6C, D	Removed	Concern over the assay (Reviewer 1); data redundancy (Reviewer 2)	Reviewer 1, 2
Figure EV2B	Figure EV2B	Endogenous MAST4 signal was unclear on immunoblots	Reviewer 1
Figure EV4	Figure EV6	Background signal, low-power view of an entire cell	Reviewer 1
-	Figure 3B	A ciliary resorption experiment was suggested to further support Tctex-1 is downstream of MAST4	Reviewer 2
-	Figure EV4	3D structure prediction of MAST4 kinase domain	Reviewer 3
-	Figure EV7	Full blots of Cdc42 activity assays are desired	Reviewer 1

Reviewer #1

1) This manuscript describes the identification of ciliary MAST4 and its role in ciliary resorption. MAST4 KD cells resorb cilia with delayed kinetics, and MAST4 overexpression accelerates population-level ciliary disassembly. This paper presents data indicating that MAST4 physically interacts with Tctex-1 and identifies the responsible protein fragment. Sakaji et al hypothesize that MAST4 phosphorylates Tctex-1, with supporting data that in MAST4 KD cells phosphomimetic Tctex-1 restores near control level population resorption kinetics. A selection of point mutants reveals the amino acids linked to defects in resorption. In the absence of MAST4, Cdc42 activation is perturbed, providing some rationale on how MAST4 fits into the larger known mechanisms underlying ciliary resorption.

Overall the manuscript was well written, provides new information about ciliary resorption, and will be interest to Life Science Alliance readers. All comments offered herein are in good spirit to both help the authors and future readers reproduce these experiments.

2) Main points:

- MAST4 accelerates ciliary resorption

Yes, the data is supportive. It would be enhanced with better MAST4-mC images as those presented in Figure 1 E-F have much background making it difficult to distinguish signal from noise (especially in E).

Reply: As suggested, we replaced the original Fig. 1E and F with a better-quality image in revised Fig. 1E and Fig. EV3, respectively, to depict the ciliary distribution of MAST4-mCherry.

According to this reviewer's suggestion below, we also added a low-power view of a transfected cell in both figures. As a result of these addition, we removed the original Fig. 1F to the Supplementary figure EV3 due to space constraints in the main figure.

- The kinase domain of MAST4 directly binds to Tctex-1 Blots in Fig 2 B are not sufficient to claim endogenous MAST4 co-immunoprecipitates with Tctex-1.

Reply: We agreed with the reviewer retrospectively. The original Fig. 2B showed that the amount of the endogenous Tctex-1 pulled down by the MAST4 antibody was low, and, hence not a strong evidence supporting the endogenous protein-protein interaction. We, therefore, deleted the original Fig. 2B.

•MAST4 enhances the ciliary transition zone expression of phospho-(T94)Tctex-1
Without details on how these experiments were performed, I cannot evaluate if the data support the claims or not.

Reply: We apologize for our oversight. We extensively rewrote the Materials & Methods section and added greater detail for the procedures for cell staining, imaging, and quantification of the ciliary base signal of phospho-(T94)Tctex-1.

•The kinase activity of MAST4 is required for Tctex-1-mediated ciliary resorption
The data supports that the kinase activity of MAST4 is required for ciliary resorption. Again, need details on qIF to interpret properly.

Reply: As suggested, we added additional detail for qIF (also see above). Retrospectively, we agreed with the reviewers that our available data are not adequate to directly support the original claim calling “the kinase activity of MAST4 is required for ciliary resorption”. We deleted that statement from the manuscript. Overall, we significantly softened the original claim. Instead of calling MAST4 is the kinase that phosphorylates Tctex-1, we concluded MAST4 is a novel kinase that regulates ciliary resorption by modulating the ciliary base activation of phospho-(T94)Tctex-1.

•MAST4 regulates ciliary resorption by activating the downstream effectors of phospho-(T94)Tctex-1
The data supports indirect activation of downstream effectors & upstream involvement of MAST4 with Tctex-1. I cannot interpret how the GFP-F experiment was quantified due to lack of details.

Reply: We agreed with this reviewer more detailed description of these experiments will help data interpretation. However, a holistic consideration of this reviewer’s comment and the comments from Reviewer 2, who called the GFP-F studies are redundant, we decided to delete the original Fig 6C, D. We believed that the removal of these figures did not impact on our conclusions.

3) Major points: For interpretation & reproducibility, this manuscript needs a M&M section on quantitative immunofluorescence for figures 3,4,6. Presumably, for Fig3-4, you've measured the intensity of the signal in the channel of interest? This should really be done with a reference channel, i.e., a known TZ marker to make an ROI for measuring your channel of interest. Did you subtract background fluorescence? "How did you take these measurements with puncta of variable size? Figure 6C-D is super unclear how one quantifies such "periciliary" signal

Reply: We rewrote the M&M by adding a new section "Immunostaining, imaging, and immunofluorescence quantification" to describe the quantitative immunofluorescence of Fig. 3, 4, and 6 in detail (note Fig. 6C and D were deleted). It now reads "For quantitative immunofluorescence analysis, the confocal fluorescence intensity of MAST4-mCherry across the entire cilium and ciliary base was plotted using Zen software (Zeiss). The representative line scans from the images of 10-16 cilia were shown. To quantify the ciliary basal signal of phospho-(T94)Tctex-1, the total fluorescence of the region of interest (ROI), i.e., the proximal end of Ac-Tub labeled cilium, was quantified in the projection confocal images of 1.25 μm -thick Z-series sections of cilium-containing regions ($x = 1 \mu\text{m}$, $y = 1 \mu\text{m}$) in randomly selected cells using NIH ImageJ 64 software. The numbers of analyzed cilia were indicated in the figure legend. All experiments were repeated at least three times. The ciliary length was measured using the acquired confocal Ac-Tub images on NIH ImageJ 64 software. For presentation purposes, the brightness and the background of the images were adjusted by Photoshop Elements 12 software (Adobe Systems)."

With due respect, we were unable to provide co-labeling of phospho-(T94)Tctex-1 with CEP290 (the most commonly used transition marker) because both antibodies were made in rabbits. The ciliary transition zone distribution of phospho-(T94)Tctex-1 was previously described by Li et al., Nat. Cell Biol., 2011. Demonstrating the ciliary transition zone location of phospho-(T94)Tctex-1 is not pertinent to the conclusion of the current report. Therefore, in the revised manuscript, we described the distribution of phospho-(T94)Tctex-1 at the ciliary base.

Figures 3 & 4: There's a puncta for P-(T94)Tctex-1 in the KD & point mutants so presumably some other kinase is phosphorylating this substrate. Please add to the discussion that MAST4 is not solely responsible (unless you have additional supportive

data to include) & anything that's known about other kinases.

Reply: We appreciate this valuable suggestion. We discussed this possibility in the revised Discussion (Lines 355-360).

Additional specific issues:

General: some wording throughout like "expression of X protein at ciliary TZ"-I think you mean localization? Surely not implying gene expression, right? I'm also unclear on how a transient interaction like that of a kinase & substrate can be stable enough for a pulldown. Some discussion might help here.

Reply: We made the editorial changes as suggested, we used either "localization" or "distribution" instead of "expression".

The reviewer is right; stable complexes are more likely to be co-immunoprecipitated. However, precedents showed certain kinases and their substrates could be identified through pull-down assays especially when these proteins are overexpressed (e.g., Dart et al., J. Biol. Chem., 2001). MAST4 was initially identified in the proteomics of the GFP immunoprecipitants of Tctex-1-GFP stably expressed in the RPE-1 cells (Saito et al., EMBO Rep., 2017). The current paper showed that MAST4-mCherry pulled down the co-expressed FLAG-Tctex-1 in RPE-1 cells (Fig. 2A). The rest pull-down assays were carried out using recombinant bacterial fusion proteins in vitro (Fig. 2C).

Figure 1 B: It doesn't say here, but I'm assuming these measurements were made off ac-tub signal. That's a bit problematic since deacetylation happens before resorption as the authors cite in Pugacheva et al.

Reply: The details of cilium length measurement was added in the revised manuscript under the newly added section "Immunostaining, imaging, and immunofluorescence quantification". We described that "The ciliary length was measured using the acquired confocal Ac-Tub images on NIH ImageJ 64 software (Lines 499-500). The reviewer is right, Ac-Tub-labeled cilium has begun to shorten before resorption. Nevertheless, Pugacheva, et al., and several subsequent studies (including our own paper Li et al., Nat. Cell Biol., 2011) suggested cilium length has a good positive correlation with the fraction of the cell display a cilium. This makes sense why we chose the ciliated cell

fraction (vs. cilium length) as the major read out for the ciliary resorption assay.

E: That's a lot of background for MAST4-mC. The fluorescence intensity line plot should be segmented to cover the length of the cilium as defined by ac-tub staining. As displayed, it looks like mC signal ends at tip, but as drawn, it looks like it's due to the straight line. If not, fix the line in the first panel to reflect appropriately. Honestly, not sure if this adds much though will all the background. What's the true signal here? Would like to see larger whole cell pic to get a better idea of signal-to-noise.

Reply: As suggested, in the revised Fig. 1E, we modified the fluorescent intensity line plot so that the Ac-Tub staining spanning the entire cilium was segmented and presented. We also better labeled the orientation (basal body at left and distal cilium at right) in the y-axis. Also, we added a whole cell view, and, understandably, MAST4-mCherry is not exclusively expressed in the cilium. The cilium-associated MAST4-mCherry signals are likely to be true signals because no similar ciliary association was found in RPE-1 cells transfected with mCherry alone (data not shown).

F: need scale bar & asterisks definition

Reply: We added the scale bar and asterisk definition in the original Fig. 1F (now revised Fig. EV3).

Figure 2

A & B: These blots leave much to be desired. With signals like these, this reviewer requests full panel of these membranes. Tctex-1 "signal" is especially problematic as it doesn't even really look like a band, but rather a smudge from signal above. Shouldn't these weights be different for MAST4-mC & endogenous MAST4? Looks nearly identical.

Reply: As suggested, the revised Fig. 2A shows a full panel of the original Fig. 2A. As shown, a clear Tctex-1 (Flag-Tctex-1) band was detected in the immunoprecipitant pulled down by mCherry antibody but not control anti-GFP antibody.

The original Fig. 2B was aimed to pull down the endogenous Tctex-1 using anti-MAST4 antibody from RPE-1 cells. Understandably, co-immunoprecipitation of endogenous (vs. overexpressed) proteins is generally more challenging due to the trace amount of

expression. Two factors further increased the practical difficulty of this experiment. (1) Transferring a very big protein (MAST4) and a very small protein (Tctex-1) simultaneously in a same protein blot, and (2) Tctex-1 and immunoglobulin light chain (used for IP) have similar molecular weights. We agreed with this reviewer that this blot leaves much to be desired. As a result, we decided to delete the original Fig. 2B considering this removal will not impact the conclusion of the present paper.

D: Signal for frag 1 too in anti-MBP blot? Why aren't we seeing a clear 56 kDA band in lane 2 indicative of MBP-Tctex-1?

Reply: We agreed with this reviewer about the first comment. In the revised manuscript, we mentioned weak signal of MBP-Tctex-1 was also pulled down by GST-MAST4 fragment 1 (Lines 181-182).

We respectfully disagreed with the second comment. We speculated this misunderstanding was due to the poor figure labeling. Therefore, we revised the labeling of the original Fig. 2D (revised Fig. 2C). We added the molecular weights of all recombinant proteins tested and marked their position on protein gels.

Figure 3

A: for consistency, please label the figure with MAST4-sh1

B: missing scale bar

Reply: As suggested, we corrected the label in Fig. 3A. We added the scale bar in original Fig. 3B (now revised Fig. 3C).

Figure 4

C: Of all the puncta in these little panels, you're quantifying those that overlap with ac-tub signal?

Reply: We quantified the phospho-(T94)Tctex-1 signals associated with the proximal end of Ac-Tub labeled cilia. One of the panels in this figure (Fig. 4C, WT 2 h) was replaced with a better-quality representative image.

Figure 5

C-F: Need a better explanation of how you're measuring active Cdc42 (I had to google

it but it should be in the paper. If you're including densitometry of blots & running stats, this reviewer wants to see full panels in the supplement of all blots used to generate D & F. It's also unclear how you're measuring relative levels: is the active & total Cdc42 on the same membrane? I should hope so. Need more in M & M than Ins 466-467.

Reply: We apologized for the overlook. As suggested, we added more details about the Cdc42 activity assay in revised Results (Lines 264-275) and M&M (Lines 573-589). The active form of Cdc42 and the total Cdc42 ran on the same protein gels and were detected by the anti-Cdc42 antibody on immunoblots. The full panels of Figs. 5D and E were presented in the revised Fig. EV7.

Figure 6

B: How are you differentiating/identifying internalized Tf-594. From this data, I cannot believe those specific puncta are internalized vs all the others.

Reply: We revised the M&M by explaining the detailed steps of the transferrin (Tf) uptake assays (Lines 516-530). Briefly, upon completing the incubation with extracellularly added Alexa-594 conjugated transferrin, the cells were transferred on ice and followed by a series of washing including a mild-acid buffer washing. This low pH allows the dissociation between Tf and Tf receptors on the cell surface and hence removes the surface-bound Tf. Previous studies using radioactive Tf showed that the acid washes remove >90% of the transferrin bound to the surface (Johnson et al., MBC 2011). This is an established protocol and has been widely used for studying the trafficking of Tf.

The ciliary pocket is a site of robust clathrin-mediated endocytosis. Robust internalized Tf signal has also been similarly observed near the ciliary base by other groups (Molla-Herman et al., J. Cell Sci., 2010; Clement et al., Cell Rep., 2013) and our previous studies (Saito et al., EMBO Rep., 2017; Yeh et al., Dev. Cell 2013).

C: This experiment seems shaky from what's presented. How are you claiming this signal is from the ciliary pocket?? See major points above regarding quantification of this experiment.

D: Figure legend says this is in %-please indicate likewise on the figure

Reply: The periciliary tubules of transfected GFP-fusion of farnesylation sequence

(GFP-F) has been used as a surrogate of ciliary pocket membranes (Molla-Herman et al., J. Cell Sci., 2010). Taken together with this reviewer's concern and Reviewer 2's concern about the redundancy of this piece of data (i.e., GFP-F fluorescence quantification), we decided to delete the original Fig. 6C and D and the related statements.

EV2 Why isn't there an endogenous MAST4 signal in B while you can see it in A? GAPDH loading is similar so one should see it.

Reply: In Fig. EV2B, the high level of transfected MAST4 is likely to obscure the relatively lower level of endogenous MAST4 due to the former sequesters the majority of the primary and secondary antibodies. Regardless, the most important message here is that the transfected MAST4 and endogenous MAST4 migrated to the similar position on protein gels.

EV3 What's going on with K506A signal here? Need to show clearly that it is expressed for other experiments.

Reply: For unclear reasons, the transfection rate of MAST4^{K506A} was lower compared to that of other variants, this was consistent with its weaker protein signal on immunoblots (Fig. EV5). Since we only counted the cilium expression in GFP⁺ transfected cells, the lower transfection rate of MAST4^{K506A} would not affect the cilium counting results. Nevertheless, for the subsequent studies (Figs. 4C, D, EV6), we primarily focused on testing MAST4^{R503A} and MAST4^{D504A} mutants. These comments were added in revised Results (Lines 231-236).

EV4 Way too much mC background here to say where this signal is coming from! One could generate line intensity plots anywhere in these small panels. This is not acceptable data to support subcellular localization. The figure isn't labeled with mutants. No scale bar definition in figure legend.

Reply: We replaced the original Fig. EV4 (revised EV6) with a better-quality representative image with lower background. We also added a whole cell view, the label of the mutants, and scale bars.

Ln 111: These are not isoforms of one protein but rather a family of kinases as you said

in the intro. This sentence leads one to believe these are isoforms of MAST (even a thing)? Consider “MAST4 is the predominant kinase of the MAST family”

Reply: Thanks for the suggestion. We described this point in the Introduction (Lines 109-110).

Ln 140: is this ciliary pocket staining? I'd like to see a secondary pocket marker to be convinced

Reply: We agreed with the reviewer calling the ciliary pocket localization of MAST4-mCherry requires the co-labeling with a marker. We avoid performing the colocalization study using the transfected ciliary pocket reporter (GFP fusion of farnesylated motif), which might cause unnecessary complications. Since the ciliary pocket localization of MAST4 is not pertinent to any key conclusion of the paper, we simply avoid calling this out. In the revised Results and figure legends (original Fig 1F/revised Fig. EV3), we described “MAST4^{WT}-mCherry signals were detected in the ciliary base (arrow) and ciliary axoneme (arrowhead). The asterisk marks the MAST4^{WT}-mCherry signal juxtaposing to the ciliary axoneme, possibly representing the ciliary pocket..” (Lines 956-958)

Ln 140-142: how do these data support that MAST4 plays an imp't role in only the early phasic resorption 0-2hr when resorption is also perturbed at 24 hours?

Reply: Thanks for the comment. We removed the “early phase” from this context in the revised Results (Lines 162-163).

Ln 144: seems like an illogical jump here if you read results first. Consider adding a similar statement about the interaction identified in a previous paper here

Reply: As suggested, we added an introduction sentence in the first paragraph of Result.

Ln 175: Haven't shown MAST4 phosphorylates Tctex-1-either provide direct data or adjust to "likely phosphorylates" or the like

Reply: We significantly softened the claim throughout the entire revised manuscript. We

added comments in the revised Discussion describing the technical challenges that prevented us from provide evidence directly supports MAST4 as a kinase that phosphorylates Tctex-1 (Lines 350-355).

Ln 181: AGC? Define

Reply: We defined abbreviation of AGC (Line 109).

Ln 201-202: cannot say from these images that anything moved from basal body to axoneme

Reply: We made the editorial correction as suggested (Lines 238-239).

Ln 203: expression→ localization? "Variants resulted in reduced signal from"

Reply: As suggested, we replaced the word of "expression" to localization in the related context.

Ln 442→ TIRF?

Reply: Thanks for the comments. We corrected "TRIF" to "TIRF" (Line 528).

Ln 704: scale bar in F also 1?

Reply: We added the scale bar in the original Fig. 1F (revised Fig. EV3).

Ln 724-725: could put kDa in the figure easier for the reader to interpret the blot below-to scale with C

Reply: We appreciate the suggestion. We labeled the expected molecular weights (KDa) of all recombinant proteins in the figure.

Ln 776: PAK-PBD beads? Please explain these beads so readers don't have to google to figure out your experiments.

Reply: As suggested, we added the rationale and procedures of using PAK-PBD beads

for Cdc42 activity assays in revised Results (Lines 267-268) and Materials & Methods (Line 581).

Ln 812: major member? Consider "predominately expressed"

Reply: We made editorial change by using the suggested, correct wording (Line 939).

Reviewer #2

1.

In this work Sakaji et al. study the phenomenon of primary cilia resorption in a cell line in culture. In particular, they address the function of MAST4 (a Ser/Thr kinase) in the process, paying special attention to its interaction with Tctex-1 as a potential substrate. Tctex-1 is a protein that was previously identified as a participant in cilia resorption. The phosphorylation of Thr94 in Tctex-1 (T94) is necessary for the activation of downstream effectors involved in ciliary resorption. In this work the authors determine that MAST4 is involved in ciliary resorption, interacts with Tctex-1 by its kinase domain and that the downstream mechanisms involved in ciliary resorption are the ones triggered by Tctex-1 phosphorylation, thus proposing Tctex-1 a substrate of MAST4. The data obtained in this work provides new information about ciliary resorption, a process that is still not completely understood. It is clear from their work that MAST4 is another player in ciliary resorption, and although they provide evidence of interaction between this kinase and Tctex-1, I think some other simple experiments would further support the claim that MAST4 activity in ciliary resorption is due to Tctex-1 phosphorylation.

2.

a) MAST4 accelerates ciliary resorption

The participation of MAST4 in ciliary resorption is clearly supported by knockdown (KD) and overexpression (OE) experiments and the evaluation of the cilia length and % of ciliated cells in transfected cell lines. The conclusion that MAST4 is localized at the ciliary pocket needs to be further confirmed using a marker of the ciliary pocket in the immunofluorescence (IF). This marker was used in Figure 6 and thus should be included in the IF.

Reply: Claiming the localization of MAST4 at ciliary transition zone and ciliary pocket localization is not pertinent to the conclusion of this report. As a result, we modified the claim by stating "MAST4-mCherry is broadly distributed in the cilium, including the ciliary base, the ciliary axoneme and the region juxtaposing to the ciliary axoneme, possibly ciliary pocket".

The co-staining of the most commonly used ciliary transition marker CEP290 is not feasible because that both CEP290 and MAST4 antibodies were made in rabbits. The colocalization study of ciliary pocket marker GFP-F (GFP fusion of farnesylated motif)

requires transfection, which might introduce additional variables.

b) The kinase domain of MAST4 directly binds to Tctex-1

The authors observed the interaction between MAST4 and Tctex-1 when they overexpressed both proteins, and also with the endogenous proteins. The WB showing the immunoprecipitation of endogenous proteins, which reflects physiological interaction, is not of very good quality, it would be great if it could be improved.

Reply: A similar comment regarding Fig. 2B was also raised by Reviewer 1. Here, we reiterated, "The original Fig. 2B was aimed to pull down the endogenous Tctex-1 using anti-MAST4 antibody from RPE-1 cells. Understandably, co-immunoprecipitation of endogenous (vs. overexpressed) proteins is generally more challenging due to the trace amount of expression. Furthermore, two additional factors increased the practical difficulty of this experiment. (1) Blot transferring a very big protein (MAST4) and a very small protein (Tctex-1) simultaneously; (2) Tctex-1 and immunoglobulin light chain (used for IP) have similar molecular weights. We agreed with this reviewer that this blot leaves much to be desired. As a result, we decided to delete the original Fig. 2B considering this removal will not impact the conclusion of the present paper."

Moreover, the authors mentioned in the introduction of the present manuscript that the interaction of MAST4 and Tctex-1 was observed in a previous paper (Saito et al. 2017), in the proteomic analysis of material immunoprecipitated with an anti-Tctex-1 antibody. I could not find this result in the cited reference. It would be nice, if possible, that the authors could include the proteomic data (the corresponding author of the present manuscript is the first author of the one involving proteomics), as another evidence of the interaction between the two proteins.

Reply: As suggested, in revised Table EV1, we presented a partial list of the proteomics data obtained from the previous studies.

The evidence that the kinase domain of MAST4 is the one involved in interacting with Tctex-1 comes from pull-down experiments where Tctex-1 and different fragments of MAST4 were expressed in bacteria. The results are convincing. However, there is a contradiction between what is described in Materials and Methods and in Legend of Figure 2D. Whereas in M&M it is said that crude extracts from bacteria expressing MAST-fragments was incubated with extracts from bacteria expressing Tctex-1, in

Legend of Figure 2D says that purified MAST4 fragments were incubated with purified Tctex-1. It is important to clarify which of the two procedures was used. In case the authors have used crude extracts, they cannot assure that the interaction between the MAST4 kinase domain and Tctex-1 is direct. Please check.

Reply: We thank the reviewer for pointing out the inconsistency. We used bacterial lysates containing the recombinant proteins rather than purified proteins, i.e., not a piece of evidence supporting direct protein-protein interaction. We modified the related comments in revised M&M, legends, and Results.

I think Figure 2D could improve very much if the pull down of the different MAST4 fragments is visualized by Western Blotting with an anti-GST antibody instead of Coomassie staining.

Reply: This is a good suggestion theoretically. However, our experiments showed that anti-GST antibody has a differential affinity to different GST fusions (and their intermediate degradative products) on protein blots. The reason is unclear, but probably related to certain protein folding and consequently epitope masking/exposing. In the revised M&M, we added more details how we quantified the GST-fusion proteins. In our hands, Coomassie blue stained gels are the most reliable methods to estimate the inputs of the GST fusion proteins. With due respect, we decided to keep the original figure that shows the Coomassie blue stained gel to show the inputs of the pull-down assays are roughly equivalent.

[Figure removed by editorial staff per authors' request]

c) MAST4 enhances the ciliary transition zone expression of phospho-(T94) Tctex-1
First of all, I do not like the use of the word "expression" when referring to localization

of a protein. Expression maybe confused with translation, that does not occur at the cilium. Please change "expression" for localization.

Reply: As suggested, we avoided using the word "expression". We used either localization or distribution instead.

The effect of MAST4 KD is reversed by the overexpression of Tctex-1T94E but not the WT or the Tctex-1T94A proteins, suggesting that Tctex-1 could be a substrate of the kinase. I think the results support the idea but other evidences are needed in order to establish that MAST4 phosphorylates Tctex-1. As the authors mentioned in the Discussion section, MAST4 is a big protein, difficult to produce recombinantly so as to test its activity on Tctex-1 directly. However, as this point is very relevant for the work, I suggest that an effort has to be made in order to have more evidence of MAST4 phosphorylating Tctex-1. I am proposing this because there are at least two experiments, that look relatively simple, that could shed light on this point:

- i) it would be very informative to see what happens with Tctex-1 phosphorylation levels in MAST4 KD cells. This could be studied by WB using the anti-phospho-Tctx-1 antibody as previously done (Saito et al, 2017). The results could further support the idea that MAST4 phosphorylates Tctex-1.
- ii) Does Tctex-1T94A suppress the acceleration of ciliary resorption caused by overexpression of MAST4?

Reply. As suggested in (ii), we added a new figure showing FLAG-Tctex-1^{T94A} suppressed the acceleration of ciliary resorption caused by co-transfected MAST4 (revised Fig. 3B).

Regarding to the (i), the currently available phospho-Tctex-1 antibody is not sensitive enough to detect the protein expressed in RPE-1 cells. As a result, we softened our claim. We no longer claimed MAST4 is a kinase that phosphorylates Tctex-1. Rather, we described MAST4 is required for the ciliary base activation of phospho-Tctex-1.

We did previously showed a WB blot of phospho-(T94)Tctex-1 in serum-stimulated RPE-1 cells (Fig. 2 in Li et al., Nat. Cell Biol., 2011). However, due to the very low level of phospho-Tctex-1, in these experiments, immunoprecipitation was first carried out using a saturated amount of anti-(pan)Tctex-1 Ab, and the immunoprecipitates were electrophoresed and immunoblotted with phospho-(T94)Tctex-1 Ab (see Fig.2 legends

in Li et al., Nat. Cell Biol., 2011). We felt strongly that this IP/IB approach is not reliable for a quantitative comparison between specimens.

MAST4 KD also decrease the serum-induced localization of phospho-Tctex-1 at the transition zone (TZ). This is clearly supported by the IF images.

There is an issue regarding Tctex-1 localization that is not clear. I did not find in the previous papers (Li et al 2011 and Saito et al 2017) which is the localization of un-phosphorylated Tctex-1. Is it also present at the TZ and then phosphorylated at this place? Or is it present somewhere else and moved to the TZ upon phosphorylation? Would you be able to answer these questions?

Reply: The ciliary localization of Tctex-1 has not been reported using the (pan)Tctex-1 antibody. Nonetheless, proteomics of primary cilia identified the presence of Tctex-1 (aka DYNLT1) along with other dynein components (Fig 2, Mick et al., Dev. Cell, 2015, PMID 26585297). Given the broad ciliary distribution of MAST4, we do not know the exact location where Tctex-1 is phosphorylated. Related discussion is not included in the revised Discussion.

d) The kinase activity of MAST4 is required for Tctex-1-mediated ciliary resorption. The results supporting that the catalytic domain of the kinase is important for ciliary resorption and phospho-Tctex-1 TZ localization are convincing.

Reply: Thank you.

e) MAST4 regulates ciliary resorption by activating the downstream effectors of phospho-(T94) Tctex-1

I consider that this conclusion is well supported by the experiments and the results.

Reply: Thank you.

However, I found an inconsistency with the previous paper (Saito et 2017). In Figure 6B there is no effect of the overexpression of the Ccdc42 CA protein, while it was expected to accelerate ciliary resorption. Please comment on that.

Reply: Fig. 5D in Saito et al. (EMBO rep., 2017) showed overexpression of Cdc42-CA accelerated ciliary resorption. Instead beginning at 2 h time point, cilia began to resorb

0.5 h and 1 h post-serum readdition in these cells. Nonetheless, the percentage of ciliated cells is not significantly different between 2 and 24 h time points in Cdc42-CA transfected cells. In the current manuscript, Fig. 5B, we only measure the 2 and 24 h time points in Cdc42-CA transfected cells. So that the acceleration of ciliary resorption is not observed, as expected (Lines 261-263).

Once the authors showed that MAST4 mediates Cdc42 activation and Arp2/3 activity, the rest of the experiments involving endocytosis of the ciliary pocket membrane are a bit redundant, as the link between Cdc42-Arp2/3-endocytosis of the ciliary pocket membrane and ciliary resorption was previously established (Saito et al 2017).

Reply: We appreciate the viewpoint. As suggested by this reviewer stating “the rest of the experiments” (after Fig. 6A &B) are redundant, we decided to delete the original Fig. 6C, D (i.e., the GFP-F ciliary pocket membrane assays). Please also see the related comments raised by Reviewer 1.

f) Discussion

Line 259-262: Our mechanical studies showed that MAST4 regulates ciliary resorption by promoting the expression localization of phospho-(T94)Tctex-1 in the ciliary transition zone and its downstream activation of Cdc42-ARPC2-mediated ciliary pocket membrane endocytosis.

"Endosomes from the ciliary pocket membrane do not simply remove the ciliary components (e.g., depolymerized tubulin, ciliary membrane) after axonemal microtubule disassembly; they would actively prime ciliary resorption". I do not understand how depolymerized tubulin, a soluble and intracellular protein would be eliminated by endocytosis.

"This idea is supported by the fact that phospho-Smad2/3 accumulates on the endosome membrane, which is generated from the ciliary pocket membrane (Saito et al., 2017)". Please expand on the link between activation of Smad signaling and ciliary resorption as it is not clear from the reference.

Reply: We wrote this part in revised Discussion. It reads, “The endosome has long been recognized as a signaling hub that couples the signals sensed by the cell surface

receptor, the distribution of intracellular signaling components, and the gene regulation in the nucleus (Scita & Di Fiore, 2010). Accordingly, the downstream signaling components of several ciliary membrane receptors are localized in the endosomes at the ciliary base. For example, phospho-SMAD2/3, the notable downstream effector of the TGF- β receptor (a ciliary membrane protein), is abundantly accumulated in the periciliary endosomes during ciliary disassembly (Clement et al., 2013; Saito et al., 2017). It remains to be investigated how the internalized surface signals modulate a cascade of cellular events that ultimately harness the process of ciliary resorption.” (Lines 323-332).

"MAST4 certainly controls corticogenesis by phosphorylating (T94)Tctex-1 in the cells". This idea has not been tested in this work, so it should be expressed in a conditional way.

Reply: As suggested, we corrected the wording in the revised Discussion (Lines 369-370).

3. Lastly, indicate any additional issues you feel should be addressed (text changes, data presentation, statistics etc.)

Pg 2 line36: change expression for localization

Pg 3, line 52: change /"cilium-expressing" for ciliary proteins.

Pg3, line 55: change "membranes" for membrane.

Pg 4, line 73: "as well" is redundant with "also" in previous line.

Pg 4, line 75: replace "ciliary" by cilium.

Pg 4, line 79: replace "act on" by act in.

Pg 4, line 85: add reference after "ciliary resorption"

Pg 5, line 98: re write "isolated as a component in the proteomics of Tctex-1..." as identified an interactor of Tctex-1 in...

Pg 5, line 106: change expression for localization.

Pg 6, line 114: change "different sequences" for different MAST4 sequences.

Pg 6, line 115: the reference Li et al 2011 should be after the word "plasmids" in the previous line.

Pg 6, line 117: move "by immunoblotting" after " We confirmed" in line 115.

Pg 6, lines 119-121: re-write "The length and number of the green fluorescent protein (GFP)+ transfected cells with cilia, labeled with anti-acetylated α -tubulin (Ac-Tub)

antibody, were analyzed" as Cilia length and percentage of ciliated cells was determined in GFP positive, transfected cells, using immunofluorescence and confocal microscopy.

Pg 6, line 125: eliminate "the"

Pg 7, line 134: replace "ciliary expression" by subcellular localization.

Pg 7, line 141: again, change "expression" for localization or presence

Pg 8, line 162: change "expression" for localization.

Pg 8, line 172: replace "depleted" by decreased.

Pg 8, line 173: eliminate "s" from "zones"

Pg 8, line 175: change "expression" for localization

Pg 11, line 231: change "extracellularly fed" by extracellular

Pg 7, line 139: again, change "expression" for localization the first time and presence the second time.

Pg 2 line 36: change expression for localization

Reply: We extensively rewrote the manuscript and make sure to avoid the same editorial mistakes.

Pg 3, line 52: what is meant by "transition hub"? "Transduction hub" maybe?

Reply: Thank you for pointing out this typo. We corrected the typo; it is "transduction hub" (Line 55).

Pg 4, line 84: specify ciliary pocket membrane components.

Reply: The original Line 84 is an editorial error. The correct sentence reads "We have previously identified Tctex-1, or dynein light chain Tctex-type 1 (DYNLT1), as a vital component for ciliary resorption (Line 92-93).

Pg 6, line 132: The last sentence does not correspond to what is shown in the figure. % of ciliated cells rather than cilia length is shown in Figure 1D. Please correct.

Reply: As suggested, we corrected the sentence. Instead of calling the cilium length, we changed it to the % of ciliated cells.

Materials and Methods

Ciliary resorption assay, GFP-F imaging assay, and immunostaining
Please provide details of the temperature of incubation of the antibodies.

Co-immunoprecipitation assay and western blotting
Please provide details of the time and temperature of incubation of the antibodies

Reply: We added more experimental details (including the antibody incubation, duration, and temperature) in the revised M&M.

Pg 30, line 707-708: Please re-write the following sentence "A representative immunoblot of HEK293 cells overexpressing MAST4WT-mCherry and FLAG-Tctex-1WT was subjected to a co-immunoprecipitation assay" as "A representative immunoblot of an immunoprecipitation assay using HEK293 cells overexpressing MAST4WT-mCherry and FLAG-Tctex-1WT".

Reply: We rewrote the entire sentence to avoid the confusion (Lines 829-832).

Pg 30, line 712: idem

Reply: We deleted the original Fig 2B as mentioned above. Thus, the sentence was also deleted.

Pg 30, line C: Diagram showing MAST4 domains

Reply: We modified the legends as suggested (Line 833).

Pg 31, line 738: Values (C)...is the same for Values in A? Please include.

Reply: We corrected the legends accordingly (Lines 844-867).

Legend of Figure 5. "The histogram shows the percentage of GFP+ cells expressing Ac-Tub-labeled primary cilia at the indicated time points". Did you count GFP+ cells or GFP+-HA+ cells?

The same question applies for Figure 6, is GFP+ cells or GFP+-FLAG+ cells?

Reply: For the cilium-displaying cell counting, we counted GFP+ cells.

For all experiments, we routinely performed co-staining of the co-transfected proteins in the second day after the transfection. These studies consistently showed that the co-transfection rate is nearly 100%.

Legend of Figure 6C: please specify the type of microscopy used (confocal, airyscan, SR-SIM)

Reply: We deleted the GFP-F data as mentioned above. Thus, the original Fig 6C was deleted.

Reviewer #3

Reviewer summary

This study follows up on previous findings that Tctex-1-in its T94 phosphorylated state and dynein independent function- regulates cilia resorption by activating cdc42 and regulating the branching dynamics of actin that lead to ciliary resorption. In order to further study the mechanisms underlying Tctex-1 function, this study builds upon the identification of MAST4 as a ligand for Tctex-1 using a proteomics approach. In this work, the authors analyze MAST4 role in ciliary dynamics as well as the physical and functional association between Tctex-1 and MAST4. The data provided support the conclusions of a physical and functional interactions. The study also aims to study the involvement of MAST4 kinase activity and does succeed in providing evidence that MAST4 kinase domains residues are important for Tctex-1 function and Tctex-1 effectors. However more evidence is needed to demonstrate that MAST4 kinase activity is involved in these processes.

Overall, the work is relevant to the field of cilia dynamics, ciliopathies and contributes original information to the understanding of primary cilium regulatory mechanisms. The manuscript is well written and is clear except for the parts where I asked for clarification (mostly in methods description)

The length of the paper is appropriate. Data and literature are succinctly but sufficiently discussed.

The major claims made by this report are that:

- 1.MAST4 is expressed in cilia
- 2.MAST4 and Tctex-1 interact physically and functionally
- 3.The catalytic loop of MAST4 is important for Tctex-1 phosphorylation
- 4.Phosphorylated TcTex-1 is localized to the transition zone of the cilium during resorption
- 5.MAST4 Kinase activity is required for Tctex-mediated cilia resorption

Claims 1-3 are sufficiently supported by the data in this manuscript.

For claims 4 and 5 additional experiments or rephrasing would be necessary to accurately reflect the conclusions allowed by the data.

The statement referred to the kinase activity of MAST4 being required for the observed that needs to be consistent throughout the manuscript. My major concern is related to the section 4 of results (the kinase activity of MAST4 is required for Tctex-1-mediated ciliary resorption). In order to fully support this statement, some direct readout for the kinase activity of MAST4 Wt and the point mutations should be provided, since a phosphoTctex1 antibody is available and is actually used in microscopy in this work a western blotting approach would add evidence, provided that the antibody works in western blot. The phrasing that describes the data currently available seems more accurate both in the figure legend and in the abstract.

Similarly, the conclusion about the localization of pTctex-1, should reflect that it is somewhat speculative (although reasonable) but colocalization with a transition zone markers would strengthen this conclusion.

Reviewer comments:

Methods

Cdc42 activity assay. Please provide more details not referring to previous paper. At least a rationale basis for the assay would be appreciated.

Reply: We added the rationale of Cdc42 activity assay in revised Results and procedures details in M&M section.

Statistics in Material and methods and figure legends: It has been discussed that presenting data with SEM does not add much meaningful information to the graphical representation. Rather, providing a visual representation of 95% Confidence interval- in addition to the statistical significance test results (p values)- would be more informative

Reply: We appreciate the suggestion. Our data were obtained by three independent experiments carried out using independent biological samples. The power analyses using p vale with SEM as presented is generally acceptable.

RT-PCR: I am not sure about why "cDNA was PCR amplified with specific primers" and was this followed by real time PCR? How do you keep this quantitative? Perhaps this is

2 different experiments? This is somewhat confusing and should be made more clear.

Reply: We clarified the confusion by modifying the M&M. Briefly, quantitative real time RT-PCR reactions were used to determine the transcript levels of four different MAST members using gene-specific primers. The relative mRNA levels of each kinase were calculated by the $2^{-\Delta\Delta Ct}$ method and calibrated with the expression level of a housing keeping gene GAPDH.

1. MAST4 accelerates ciliary resorption

Minor comment on Fig 1A-C: control cell resorption in biphasic manner. What does biphasic mean in this context? Could this be explained or rephrased? I am not sure that the dynamics seen Both in A and B is biphasic. It looks more linear. The rest of these panels does support the description in the text.]

Reply: In the revised Introduction, we described the serum induced biphasic cilium resorption. Briefly, this concept was first described by Golemis and her colleagues (Pugacheva et al, 2007) who used a systematic kinetic studies that showed the disassembly of the primary cilium occurs in a biphasic manner, with the first wave occurring at 2 h and a second wave at 24 h post-serum-readdition. This observation was reproduced by several other labs, including ours. We do acknowledge that the biphasic pattern is not as apparent when only two time points were tested.

Line 132; the sentence "These cilia underwent further shortly after 1 h up to 24 h time points" is perhaps mistyped. A word seems to be missing after "further"

Reply: Thank you. We revised and completed the sentence.

Fig 1E. Since there is not any good antibody available to test endogenous distribution of MAST4 , a different MAST4 fusion protein with a different tag could be used to verify the observed localization. Also, are these cells stably expressing MAST4-mCherry or are they transiently transfected?

Reply: We used transient transfection (revised Materials & Methods). The molecular cloning of MAST4, which is a large molecule (~ 265 kDa), is not entirely technically trivial. We therefore respectfully deferred the suggested experiment.

Fig 1F. Asterisks. If the picture shows a single cilium then, the bottom asterisks seem to point to the ciliary pocket but it is unclear what is signaled by the top asterisk

Reply: To avoid the confusion, we modified the labels of the original Fig. 1F (now revised Fig. EV3) and specify them in the legends.

Overall data do support the conclusion that MAST4 plays a role in ciliary resorption. The main weakness lies in the interpretation of the localization data.

Reply: This comment is well received. Taken together with the comments raised by other reviewers, we made two major editorial changes (1) we used "localization" or "distribution" instead of "expression". (ii) Because we do not have the colocalization data using a ciliary transition marker or ciliary pocket marker, we modified our description. i.e., we used "ciliary base" instead of "ciliary pocket or ciliary transition zone".

2. The kinase domain of MAST4 directly binds to Tctex-1

Fig 2A and 2B suggestion: adding IP mCherry and IP MAST4 labels to the figures would help the reading.

Reply: We revised the label of Fig. 2A as suggested. Of note, we deleted Fig. 2B in response to the comments of the other two Reviewers.

Fig 2D. Possible discussion point: the faint band detected for the pull down of MBP-Tctex-1 with Fr.1 might suggest that other regions of the protein contribute to the interaction.

Reply: As suggested, we added this comment in the revised Result section.

3. MAST4 enhances the ciliary transition zone expression of phospho-(T94)Tctex-1

The title of this section is a more accurate description of the conclusion supported by the data than the title of Figure 3 legend.

Fig 3A strong data of the functional association and the proposal of MAST4 upstream of phospho-Tctex-1

Fig 3B and C. Claiming that phosphoTctex-1 localizes at the transition zone might

require a marker for the transition zone (perhaps CEP290). Otherwise the description of the data would be more accurately described by analyzing Tctex-1 localization at the pericentriolar region or cilia base. Some of these concerns are somewhat addressed in figure EV4. A similar approach could be performed for MAST4^{WT}-mCherry. Also, these data do not fully support the conclusion that MAST4 phosphorylates TcTex1 as stated on Figure 3 legend. The title of the section is more accurate.

Reply: This comment is well received. We agreed with this reviewer that calling MAST4 phosphorylates Tctex-1 requires additional experiments (see Revised Discussion). We removed this specific claim throughout the entire manuscript including the legends of the original Fig. 3.

The ciliary transition zone location of phospho-(T94)Tctex-1 was established based on a previous paper (Li et al., Nat. Cell Biol., 2011, PMID: 21394082). The definition was also based on the topographic location (i.e., between the ciliary axoneme and basal body) rather than the co-staining with other transition zone marker. The co-staining of the most commonly used transition marker CEP290 is not feasible; both CEP290 and phospho-(T94)Tctex-1 antibodies were made in rabbits. As mentioned above, in the revised manuscript, we describe MAST-mCherry and phospho-(T94)Tctex-1 are localized at the ciliary base instead.

4. The kinase activity of MAST4 is required for Tctex-1-mediated ciliary resorption
The prediction of the catalytic motifs of MAST4 kinase domain is based on data about MAST2 kinase domain and it is correct in my opinion. Since AlphaFold2 tool is available it would be interesting to see how these predictions map on a predicted structure of MAST4. This would probably have only a visualization value since the functional data shown in 4A is sufficient for the purpose of suggesting that the catalytic activity of MAST4 kinase domain is required for promoting the resorption of cilia.

Reply: As suggested, we used AlphaFold2 tool to predict the 3D structure of MAST4 (revised Fig. EV4). The prediction shows that the R503, D504, and K506 residues are located in the highly-flexible region, likely favorable for the transfer γ -phosphate of ATP.

Line 284. Explanation for lack of direct evidence for phosphorylation of Tctex-1 by MAST4. However some experiment could be attempted. Precisely a list of potential readouts for MAST4 kinase activity is shown next.

Is there any other known substrate for MAST4 kinase activity that could be used to corroborate that the mutants have a different activity? This could be performed by a western blot of transfected cells using a phospho-specific antibody (eg phosphoTcTex-1, phosphoERM or overall levels of p-Ser) Otherwise since no direct evidence for kinase activity is shown the title of this section and the conclusion might need to reflect that the mutated residues are required for ciliary resorption, rather than the kinase activity.

Reply: We appreciated the suggestions. Also, see the replies above and to other reviewers, we removed the conclusion that “MAST4 is a kinase that phosphorylates Tctex-1” throughout the entire manuscript. Also, we added a section in the revised Discussion explaining why we were unable to perform the suggested experiments (Lines 349-360). It reads: “The above studies tempted us to propose MAST4 is the kinase that phosphorylates Tctex-1. The direct evidence supporting this claim, however, remains lacking due to several technical reasons. First, the currently available antibody is not sensitive enough to detect the phospho-(T94)Tctex-1 on immunoblots. Second, probably due to its large size, it is challenging to synthesize full-length MAST4 protein in E. coli. to perform the phosphorylation assay using the purified Tctex-1 as a substrate. Also, we cannot rule out the existence of another kinase(s) that phosphorylates Tctex-1 and this might contribute to the basal-level ciliary base signal of phospho-(T94)Tctex-1 in MAST4-KD cells. Phospho-(T94)Tctex-1 is concentrated at the ciliary base, whereas MAST4 is broadly distributed along the cilium. The ciliary location(s) where Tctex-1 is phosphorylated, either by MAST4 or other yet-to-be-identified kinases, remains unknown.”

5. MAST4 regulates ciliary resorption by activating the downstream effectors of phospho (T94)Tctex-1

Fig 5A supports the claims in the text.

An explanation of the cdc42 activity assay would be helpful for the interpretation of this article as a stand-alone set of results.

Fig 6 supports the conclusion that MAST4 acts upstream of known effectors of Tctex-1

Reply: We added the rationale and detailed procedures of Cdc42 activity assays in the revised Results and M&M sections.

Discussion

Line 260. Mechanical might not be the best word here

Line 263 where it says were it should be was

Reply: We re-wrote the first paragraph of the revised Discussion and corrected some editorial errors.

Line 303. "MAST4 certainly controls corticogenesis by phosphorylating (T94)Tctex-1 in the cells" No reference is given so this is a speculation, albeit reasonable, based on data from this paper so it should be stated as such.

Reply: We editorially corrected the statement in the revised Discussion (Lines 369-370).

The conclusions are supported partially by data. MAST4 is a regulator of cilia dynamics is supported by the data in this study. However, there is only indirect evidence showing that MAST4 kinase activity is required and that MAST4 phosphorylates Tctex-1. Since there is an anti-phosphoTctex antibody available that was used in IF, it might be possible to use it in a western blot testing the levels of pTctex-1 in the presence of each of the point mutations.

Reply: As suggested, we significantly softened the claim and no longer call "MAST4 is a kinase that phosphorylates Tctex-1" in the revised manuscript. Please see the replies above that details the reasons why we did not conduct the suggested experiments, with due respect.

Issues to be addressed:

I explained the details in previous sections but I summarize main concerns here:

Please provide direct evidence of kinase activity of MAST4 wild type and mutants preferably on Tctex-1 or on any of its known substrates. Alternatively, adjust text (mostly title and title section 4 of results and figure 3 title legend) to reflect the data available. The conclusions are more accurately described in the title of the legend for

figure 4 and result section 3.

Please provide evidence of colocalization of Phospho-Tctex-1 with a marker for the transition zone of the cilium. Alternatively, adjust text to reflect the data available.

Referee cross comments: In addition to my previous comments, I agree with the important points raised by both the other reviewers about changing the text of section 3 of results using Localization instead of expression.

Reply: Thank you for the summary and crosses reference to all the key points mentioned above. Please see the detailed replies above.

August 23, 2023

RE: Life Science Alliance Manuscript #LSA-2023-01947-TR

Dr. Masaki Saito
Teikyo University
Department of Molecular Physiology and Pathology, School of Pharma-Sciences
2-11-1 Kaga
Itabashi-ku, Tokyo 173-8605
Japan

Dear Dr. Saito,

Thank you for submitting your revised manuscript entitled "MAST4 promotes primary ciliary resorption through phosphorylation of Tctex-1". We would be happy to publish your paper in Life Science Alliance pending final revisions necessary to meet our formatting guidelines.

- please address the final Reviewer 2's points
- LSA allows supplementary figures and tables, but not EV Figures and tables; please update your callouts for the Supplementary Figures and tables in the manuscript Fig EV1A = Fig S1A, Table EV1 = Table S1)
- please add ORCID ID for the secondary corresponding author--they should have received instructions on how to do so
- please add the Twitter handle of your host institute/organization as well as your own or/and one of the authors in our system

A. FINAL FILES:

B. MANUSCRIPT ORGANIZATION AND FORMATTING:

Sincerely,

Reviewer #1 (Comments to the Authors (Required)):

This reviewer suggests acceptance of this paper as resubmitted. The authors did a thorough job addressing the comments. The new figure panels, both microscopy & WBs, look much clearer & easily interpretable. The additional M&M will be extremely helpful for readers that may attempt similar experiments.

Reviewer #2 (Comments to the Authors (Required)):

The authors had answered all the comments I made in the first round of revision. Answers are satisfactory, and when suggested experiments could not be carried out, they gave reasonable explanations. Modifications in the text were introduced to make it clearer and to soften some assertions that were not fully supported by the results obtained. I think that the revised version is clearly better than the first one and should be accepted for publication after a thorough revision of the English.

Minor comments

Ln 32: replace expression by localization.

Ln 39: replace activation by localization. Do you have evidence that phosphorylation activates Tctex-1?

Ln 44: idem to Ln 39.

Ln 113: replace composes by comprises or write ..it is composed of..

Ln 115: replace including by in.

Ln 139-140: replace ..cells expressed a cilium by ciliated cells.

Ln 140-142: Change Both measurements showed the control cells expressing GFP alone underwent biphasic resorption at 2 and 24 h after serum readdition (Figs 1A-C), as expected

By Both measurements showed, as expected, that control cells expressing GFP alone underwent biphasic resorption at 2 and 24 h after serum readdition (Figs 1A-C)

Ln 149: replace to by according to..

Ln 183: I feel it would be better if you replace binds with interacts.

Ln 188: replace involves by is involved.

Ln 198: eliminate was

Ln 201-203: I'd rather eliminate through the ciliary base activation of phospho-(T94)Tctex-1 because: i) there is no evidence that phosphorylation of Tctex-1 activates a function (you cannot discard that the stimulation of resorption is due to an inhibition of Tctex-1 function), ii) there is no direct evidence that either a) MAST4 phosphorylates Tctex-1 at the ciliary base or b) stimulates the localization of phospho-Tctex-1 to the ciliary base.

Ln 209: replace these by potential
Ln 210: replace to by with
Ln 233: replace of by in
Ln 245: idem than Ln 39
Ln 269: eliminate were
Ln 338: change recombinant proteins for extracts containing recombinant proteins.
Ln 476: change containing for in.
Ln 543: I imagine that you incubated with the antibody plus Prot G-agarose? This is not specified.
Ln 555: eliminate purification
Ln 820-821: what do you mean by high or low power view? Low of high magnification?
Ln 833: replace and by or
Ln 834: eliminate was
Ln 838-839: replace GST alone or GST MAST4 fragments for bacterial extracts containing GST or GST-MAST4 fragments.
Ln 841: What you stained with Coomassie, are the inputs or the eluates?
Ln 846: maybe is better to replace activation of the ciliary base signal of phospho-(T94)Tctex-1 by the appearance of phospho-(T94)Tctex-1 at the ciliary base.
Ln 857-858: change ..of cilium-expressing GFP+ cells by ..of ciliated GFP-expressing cells or ciliated GFP+ cells.
Ln 879: replace and by or the
Ln 880: idem than Ln 857
Ln 898: Eliminate The cells were subjected to a ciliary resorption assay, as it was said at the beginning of the legend
Ln 899: replace expressing by with
Ln 906: idem tan Ln 857
Ln 912: replace in by from
Ln 919: eliminate and
Ln 930: eliminate The cells were subjected to a ciliary resorption assay, as has been said before.
Ln 931: idem to Ln 899

Reviewer #3 (Comments to the Authors (Required)):

This study provides new data about ciliary dynamics and the mechanisms that regulate this process. It identifies the kinase MAST4 as a regulator of cilia length and supports this claim by providing convincing data showing the localization of MAST4 in the primary cilium, the physical and functional interaction of MAST4 and Tctex-1, the effect of point mutations in the catalytic domain of MAST on cilia resorption and the localization of phosphorylated Tctex-1 at the base of the cilia. After the initial revision, the paper received the editorial adjustments that were requested and justified the experimental difficulties posed by some alternative experiments. These were mostly consistent among referees on the main points about the lack of direct evidence for the phosphorylation of Tctex-1 by MAST4 and the use of the term localization instead of expression for Phospho-Tcetx1. Therefore, I recommend the publication of this paper without any further revision.

Point-by-point Responses

Editor

-please address the final Reviewer 2's points

Reply: We addressed all Reviewer2's points and made all suggested editorial modification.

-LSA allows supplementary figures and tables, but not EV Figures and tables; please update your callouts for the Supplementary Figures and tables in the manuscript Fig EV1A = Fig S1A, Table EV1 = Table S1)

Reply: We changed accordingly.

-please add ORCID ID for the secondary corresponding author--they should have received instructions on how to do so

Reply: The ORCID ID for the secondary corresponding author, Dr. Ching-Hwa Sung, is 0000-0002-3468-5867. We will fill out it in the submission procedure.

-please add the Twitter handle of your host institute/organization as well as your own or/and one of the authors in our system

Reply: N/A. My institute/organization does not have Twitter handle.

Reviewer #2

The authors had answered all the comments I made in the first round of revision. Answers are satisfactory, and when suggested experiments could not be carried out, they gave reasonable explanations. Modifications in the text were introduced to make it clearer and to soften some assertions that were not fully supported by the results obtained.

I think that the revised version is clearly better than the first one and should be accepted for publication after a thorough revision of the English.

Reply: We thank this reviewer for his/her thoughtful suggestions. We made all editorial changes as suggested. Due to the indicated line numbers being changed during the revision, we copied/pasted the revised sentences in the reply below. We underlined the changes. We sent a clean copy for revision.

Minor comments

Ln 32: replace expression by localization.

Reply: We modified as suggested. It now reads “Previous studies showed that the localization of phosphorylated Tctex-1 at Thr94 (T94) at the ciliary base critically regulates ciliary resorption by accelerating actin remodeling and ciliary pocket membrane endocytosis.”

Ln 39: replace activation by localization. Do you have evidence that phosphorylation activates Tctex-1?

Reply: We modified as suggested. It now reads “The ciliary resorption and the ciliary base localization of phospho-(T94)Tctex-1 are blocked by the knockdown of MAST4 or the expression of the catalytic-inactive site-directed MAST4 mutants.”

Ln 44: idem to Ln 39.

Reply: We modified as suggested. It now reads “These results support that MAST4 is a novel kinase that regulates ciliary resorption by modulating the ciliary base localization of phospho-(T94)Tctex-1. MAST4 is a potential new target for treating ciliopathies causally by ciliary resorption defects.”

Ln 113: replace composes by comprises or write ..it is composed of..

Reply: We modified as suggested. It now reads “MAST4 is a large protein (~265 kDa); it comprises 2,434 amino acids and encompasses a DUF1908, a serine/threonine kinase, and a post-synaptic density-95/disks-large/zonula occludens-1 (PDZ) domain.”

Ln 115: replace including by in.

Reply: We modified as suggested. It now reads” MAST4 has several reported functions in neuroprotection, spermatogenesis, the progression of multiple myeloma bone disease, and cell fate determination of mesenchymal stromal cells in bone and cartilage.”

Ln 139-140: replace ..cells expressed a cilium by ciliated cells.

Reply: We modified as suggested. It now reads “Confocal microscopic examination of acetylated α -tubulin (Ac-Tub)-labeled cilium was used to determine the length of cilia in the GFP⁺ transfected cells and the percentage of GFP⁺ ciliated cells.”

Ln 140-142: Change Both measurements showed the control cells expressing GFP alone underwent biphasic resorption at 2 and 24 h after serum readdition (Figs 1A-C), as expected By Both measurements showed, as expected, that control cells expressing GFP alone underwent biphasic resorption at 2 and 24 h after serum readdition (Figs 1A-C)

Reply: We modified as suggested. It now reads “Both measurements showed, as expected (Li et al., 2011; Pugacheva et al., 2007; Saito et al., 2017), that control cells expressing GFP alone underwent biphasic resorption at 2 and 24 h after serum readdition (Figs 1A-C).”

Ln 149: replace to by according to..

Reply: We modified as suggested. It now reads “Immunoblots showed that MAST4 (MAST4^{WT}) expressed in RPE-1 cells migrated according to the expected molecular weight (Fig S2B).”

Ln 183: I feel it would be better if you replace binds with interacts.

Reply: We modified as suggested. It now reads “These results suggest that Tctex-1 interacts with the kinase domain of MAST4.”

Ln 188: replace involves by is involved.

Reply: We modified as suggested. It now reads “The above studies prompted us to hypothesize that MAST4 is involved in the phosphorylation of Tctex-1 and, in turn, controls the phospho-(T94)Tctex-1-mediated ciliary resorption.”

Ln 198: eliminate was

Reply: We modified as suggested. It now reads “Previous studies showed that phospho-(T94)Tctex-1 prominently appeared at the ciliary base 2 h post-serum readdition (Li et al., 2011).”

Ln 201-203: I'd rather eliminate through the ciliary base activation of phospho-(T94)Tctex-1 because: i) there is no evidence that phosphorylation of Tctex-1 activates a function (you cannot discard that the stimulation of resorption is due to an inhibition of Tctex-1 function), ii) there is no direct evidence that either a) MAST4 phosphorylates Tctex-1 at the ciliary base or b) stimulates the localization of phospho-Tctex-1 to the ciliary base.

Reply: We modified as suggested. It now reads "These results collectively suggest that MAST4 acts upstream of the phospho-(T94)Tctex-1-mediated ciliary resorption through the ciliary base localization of phospho-(T94)Tctex-1."

Ln 209: replace these by potential ; Ln 210: replace to by with

Reply: We modified as suggested. It now reads "We predicted potential residues central to the kinase activity of MAST4 based on the analogy with two other AGC family members, PKA catalytic subunit α (PKACA) and human MAST2."

Ln 233: replace of by in

Reply: We modified the manuscript accordingly. It now reads "For unclear reasons, the transfection rate of MAST4^{K506A} was lower compared to that of other variants, this was consistent with its weaker signal in immunoblots (Fig S5)."

Ln 245: idem than Ln 39

Reply: We modified as suggested. It now reads "These results, taken together, suggest that the R503 and D504 residues in the kinase domain of MAST4 are required for the serum-induced ciliary resorption, likely through modulating the ciliary base localization of phospho-(T94)Tctex-1."

Ln 269: eliminate were

Reply: We eliminated "were". It now reads "The amount of the activated Cdc42 pulled down by the PBD-PAK beads and the total Cdc42, both detected by the quantitative Cdc42 immunoblotting assays, were compared (Fig 5C-F, S7)."

Ln 338: change recombinant proteins for extracts containing recombinant proteins.

Reply: We modified as suggested. It now reads "The present study experimentally validates the physical interaction between MAST4 and Tctex-1 using co-

immunoprecipitation of HEK293 transfected proteins and pull-down assays using bacterial extracts containing recombinant proteins.”

Ln 476: change containing for in.

Reply: We modified the manuscript accordingly. It reads “For immunostaining, cells grown on coverslips were fixed by 4% PFA in PBSc/m (PBS plus 0.2 mM Ca²⁺ and 2 mM Mg²⁺), quenched (50 mM NH₄Cl, 10 min), blocked (0.5% BSA, 0.01% Tritoin X-100, DAPI in PBSc/m 30 min), and incubated with primary antibodies for 1 h at room temperature in blocking buffer.”

Ln 543: I imagine that you incubated with the antibody plus Prot G-agarose? This is not specified.

Reply: The reviewer is right. We added a sentence to clarify the procedure. It now reads “The cleared supernatant containing equal amounts of proteins were incubated with 1 µg of mouse anti-mCherry antibody (or mouse anti-GFP antibody for control) for overnight at 4°C. Pre-cleared protein G Sepharose was then added for an additional 1 h at 4°C. Afterwards, protein G Sepharose was washed three times with cold STET buffer, once with STE buffer, and eluted with Laemmli sample buffer (62.5 mM Tris-HCl, 2% SDS, 5% 2-mercaptoethanol, 10% glycerol, and 0.5 mg/ml bromophenol blue, pH 6.8).”

Ln 555: eliminate purification

Reply: We modified as suggested. It now reads “Recombinant protein and pull-down assay”.

Ln 820-821: what do you mean by high or low power view? Low of high magnification?

Reply: We modified as suggested. It now reads “Scale bars: 10 µm (E, low-power magnification), and 1 µm (E, high-power magnification).”

Ln 833: replace and by or

Ln 834: eliminate was

Reply: We modified as suggested. It now reads “A representative immunoblot of total HEK293 cell lysates containing transiently transfected MAST4^{WT}-mCherry and FLAG-

Tctex-1^{WT} (input) and the immunoprecipitants (IP) pulled down by anti-mCherry or anti-GFP (control) antibodies probed with anti-RFP and anti-DYKDDDDK (FLAG) antibodies.

Ln 838-839: replace GST alone or GST MAST4 fragments for bacterial extracts containing GST or GST-MAST4 fragments.

Reply: We modified as suggested. It now reads "MBP-Tctex-1 containing bacterial lysates were mixed with bacterial extracts containing GST or GST-MAST4 fragments (with indicated molecular weights), and then incubated with glutathione beads."

Ln 841: What you stained with Coomassie, are the inputs or the eluates?

Reply: We stained GST or GST-MAST4 fragments in the elutes by CBB, as described in M&M. It now reads "The eluates from the glutathione beads were electrophoresed and immunoblotted (IB) with anti-MBP antibody, or GST or GST-MAST4 fragments in the elutes were stained by Coomassie Brilliant blue (CBB)."

Ln 846: maybe is better to replace activation of the ciliary base signal of phospho-(T94)Tctex-1 by the appearance of phospho-(T94)Tctex-1 at the ciliary base."

Reply: We modified the manuscript accordingly. It now reads "MAST4 regulates Tctex-1-mediated ciliary resorption and the appearance of phospho-(T94)Tctex-1 at the ciliary base."

Ln 857-858: change ..of cilium-expressing GFP+ cells by ..of ciliated GFP-expressing cells or ciliated GFP+ cells.

Reply: We modified the manuscript accordingly. It now reads "The histogram shows the percentage of ciliated GFP⁺ cells at 0 and 1 h post-serum readdtion"

Ln 879: replace and by or the

Reply: We modified the manuscript accordingly. It now reads "Ciliary resorption assay of RPE-1 cells transfected with GFP alone (control), or GFP together with MAST4^{WT} or the indicated MAST variants."

Ln 880: idem than Ln 857

Reply: We modified the manuscript accordingly. It now reads "The histogram shows the percentage of ciliated GFP⁺ cells at different time points after serum readdtion."

Ln 898: Eliminate The cells were subjected to a ciliary resorption assay, as it was said at the beginning of the legend

Ln 899: replace expressing by with

Reply: We modified as suggested. It now reads "Ciliary resorption assay of RPE-1 cells transfected with GFP alone (control), MAST4^{WT}, together without (control) or with HA-tagged Cdc42^{T17N} (Cdc42-DN). The histogram shows the percentage of GFP⁺ cells with Ac-Tub-labeled primary cilia at the indicated time points."

Ln 906: idem tan Ln 857

Reply: We modified the manuscript accordingly. It now reads "The histogram shows the percentage of ciliated GFP⁺ cells at different time points post-serum readdtion."

Ln 912: replace in by from

Reply: We modified the manuscript accordingly. It now reads "Representative Cdc42 immunoblots containing total Cdc42 and active Cdc42 pulled down by PAK-PBD beads from RPE-1 cells transfected with pCAG (control) and pCAG-MAST4-sh1 (C) or MAST4 variants (E)."

Ln 919: eliminate and

Reply: We eliminated "and". It now reads "MAST4 regulates periciliary membrane endocytosis working model"

Ln 930: eliminate The cells were subjected to a ciliary resorption assay, as has been said before.

Ln 931: idem to Ln 899

Reply: We modified as suggested. It now reads "Ciliary resorption assay of RPE-1 cells transfected with GFP alone (control) or MAST4-sh1-IRES-GFP (MAST4-sh1), together with FLAG-tagged Rab5^{Q79L} (Rab5-CA). The histogram shows the percentage of GFP⁺ cells with Ac-Tub-labeled primary cilia at the indicated time points."

August 28, 2023

RE: Life Science Alliance Manuscript #LSA-2023-01947-TRR

Dr. Masaki Saito
Teikyo University
Department of Molecular Physiology and Pathology, School of Pharma-Sciences
2-11-1 Kaga
Itabashi-ku, Tokyo 173-8605
Japan

Dear Dr. Saito,

Thank you for submitting your Research Article entitled "MAST4 promotes primary ciliary resorption through phosphorylation of Tctex-1". It is a pleasure to let you know that your manuscript is now accepted for publication in Life Science Alliance. Congratulations on this interesting work.

DISTRIBUTION OF MATERIALS:

Again, congratulations on a very nice paper. I hope you found the review process to be constructive and are pleased with how the manuscript was handled editorially. We look forward to future exciting submissions from your lab.

Sincerely,
